# Efficient and reproducible generation of human iPSC-derived cardiomyocytes and cardiac organoids in stirred suspension systems

Maksymilian Prondzynski [1], Paul Berkson[1], Michael A. Trembley[1], Yashasvi Tharani[1], Kevin Shani[2], Raul H. Bortolin[1], Mason E. Sweat [1], Joshua Mayourian[1], Dogacan Yucel[1], Albert M. Cordoves[2], Beatrice Gabbin [1], Cuilan Hou[1,3], Nnaemeka J. Anyanwu [2], Farina Nawar[1], Justin Cotton [1], Joseph Milosh[1], David Walker[1], Yan Zhang[1], Fujian Lu [1], Xujie Liu[1,4], Kevin Kit Parker [2,5,6], Vassilios J. Bezzerides [1] & William T. Pu [1,6] ✉

Human iPSC-derived cardiomyocytes (hiPSC-CMs) have proven invaluable for cardiac disease modeling and regeneration. Challenges with quality, inter-batch consistency, cryopreservation and scale remain, reducing experimental reproducibility and clinical translation. Here, we report a robust stirred suspension cardiac differentiation protocol, and we perform extensive morphological and functional characterization of the resulting bioreactor-differentiated iPSC-CMs (bCMs). Across multiple different iPSC lines, the protocol produces 1.2E6/mL bCMs with ~94% purity. bCMs have high viability after cryo-recovery (>90%) and predominantly ventricular identity. Compared to standard monolayer-differentiated CMs, bCMs are more reproducible across batches and have more mature functional properties. The protocol also works with magnetically stirred spinner flasks, which are more economical and scalable than bioreactors. Minor protocol modifications generate cardiac organoids fully in suspension culture. These reproducible, scalable, and resource-efficient approaches to generate iPSC-CMs and organoids will expand their applications, and our benchmark data will enable comparison to cells produced by other cardiac differentiation protocols.

Numerous cardiac differentiation protocols have been established to differentiate human induced pluripotent stem cells (hiPSCs) cultured in adherent monolayer[1,2] or three dimensional (3D) suspension[3–7] formats. However, generation of high quantity and quality human iPSC-derived cardiomyocytes (hiPSC-CMs) with low functional variability between differentiations has remained challenging. Specific reasons include insufficient quality assessment of hiPSCs[8], variability in cardiac differentiation outcomes[1,4], and loss of hiPSC-CM functional properties

[1]Department of Cardiology, Boston Children's Hospital, Boston, MA 02115, USA. [2]Disease Biophysics Group, John A. Paulson School of Engineering and Applied Sciences, Harvard University, Boston, MA 02134, USA. [3]Department of Cardiology, Shanghai Children's Hospital, School of Medicine, Shanghai Jiao Tong University, Shanghai 200062, China. [4]Fuwai Hospital, Chinese Academy of Medical Science, Shenzhen, Shenzhen, Guangdong Province 518057, China. [5]Wyss Institute for Biologically Inspired Engineering, Harvard University, Boston, MA 02115, USA. [6]Harvard Stem Cell Institute, Cambridge, MA 02138, USA. ✉e-mail: william.pu@cardio.chboston.org

following cryopreservation[9]. This variability hinders rigor and reproducibility and reduces translational value.

Presently, most cardiac differentiation methods are based on programmed activation and then inhibition of the Wnt signaling pathway[2] (Supplementary Table 1). Due to its relative simplicity and high efficiency, this strategy became the dominant approach for hiPSC-CM differentiation for disease modeling and therapeutic cardiomyocyte replacement[10–14]. Typically performed in monolayer adherent cultures, this method has become the *de facto* standard for hiPSC-CM differentiation. Nevertheless, it has a number of important limitations. Monolayer cultures scale poorly, with culture plate area and labor scaling linearly with cell number. Both cell density and environment are critical parameters for efficient differentiation. Lack of flow within culture wells results in sub-optimal distribution of nutrients and pH buffering capacity[1]. In addition, cell seeding density, a critical parameter for successful differentiation, varies locally in monolayer cultures. Together, local heterogeneity in cell seeding and microenvironment increase well-to-well and batch-to-batch variation in hiPSC-CM differentiation. While cryopreservation of cells is a critical step that dramatically facilitates design and execution of experiments and clinical translation, cryopreservation of monolayer-differentiated hiPSC-CMs (mCMs) has been reported to negatively impact their contraction, electrophysiology, and drug responses[9]. The limited scalability, well-to-well variation, and reported functional impact of cryopreservation combine to create significant barriers to the use of hiPSC-CMs.

To circumvent these limitations, suspension and stirred bioreactor cardiac differentiation protocols have been developed[3–7]. hiPSC-CMs obtained from suspension culture differentiation have been successfully utilized to model cardiac diseases[15–17] or test novel molecular therapeutics[18]. Previous studies yielded 0.5–2 million hiPSC-CMs per mL for cultures ranging from 2.5 to 1000 mL[1,3,4] (Supplementary Table 1), illustrating the scalability of suspension cultures. However, different iPSC lines exhibited inconsistent differentiation in previously reported suspension culture protocols[4]. Moreover, the morphological, contractile, and electrophysiological properties of suspension hiPSC-CMs have not been systematically evaluated and compared to monolayer-differentiated cardiomyocytes (mCMs), although metabolic analyses suggested greater maturation of suspension culture hiPSC-CMs[19].

Besides generation of hiPSC-CMs to study cardiac-specific disease mechanisms, human heart organoids[20], cardioids[21] and multilineage organoids that recapitulate cooperative cardiac and gut development[22,23] have recently significantly advanced modeling of tissue interactions in embryonic heart development. Cardiac organoids consist of self-assembled hiPSCs formed by cell aggregation and treated with defined factors that direct lineage commitment. These organoids model aspects of embryonic heart structure, spatiotemporal patterning in early cardiogenesis, congenital cardiac malformations, and regeneration after injury[20–23]. Reported organoid models are generated in static culture in multiwell dishes yielding 1 organoid per well[20–24].

Here we report an optimized suspension culture cardiac differentiation protocol with defined benchmarks enabling applicability to a variety of patient-, gene-edited and commercially available hiPSC lines. We obtained large numbers of hiPSC-CMs with high purity and developed protocols for robust cryopreservation and recovery. We extensively characterized their cellular composition and morphological and physiological properties, which collectively suggest that the suspension cultured cells have improved physiological properties compared to mCMs. Finally, we describe the first cardiac organoid (bCO) model completely generated in suspension culture. Similar to previously published cardiac organoid models generated in static culture[20,21], bCOs mainly consist of CMs and model ventricular wall and chamber formation.

## Results

### Optimized bioreactor differentiation protocol

We modified bioreactor and suspension cardiac differentiation protocols[4,5] with goals of improving yield and reproducibility, increasing applicability across diverse hiPSC lines, reducing cost, and enabling cryopreservation and recovery of resulting hiPSC-CMs. Our optimized workflow (Fig. 1a and Supplementary Fig. 1a) built on previously described embryoid body suspension culture protocols[4,5] by incorporating the following features: (1) use of quality-controlled master cell banks (MCBs) to ensure consistency of input hiPSCs; (2) use of a stirred bioreactor that continuously monitors and adjusts temperatures, O2, $CO_2$, and pH; (3) use of small molecules rather than growth factors, which are more expensive and vulnerable to lot-to-lot variation, to guide differentiation; (4) optimization of the time point to initiate differentiation by Wnt activation; (5) optimization of the duration of Wnt activation and the timing of Wnt inhibition; and (6) incorporation of controlled freeze and thaw protocols.

High quality input hiPSCs are critical for successful and consistent differentiation[25]. Towards that end, we implemented procedures to establish MCBs of quality-controlled hiPSCs (see Methods), including karyotyping (Supplementary Fig. 1b) and mycoplasma testing. To monitor undifferentiated status of hiPSCs input into the differentiation protocol, we measured pluripotency marker SSEA4 by FACS. High differentiation efficiencies (>90% expressing cardiomyocyte marker cardiac troponin T [TNNT2]) were correlated with high SSEA4 ( > 70%) values, and low SSEA4 ( < 70%) predetermined failed differentiation (<90% TNNT2+; Fig. 1b; Supplementary Fig. 1c).

In suspension culture, hiPSCs spontaneously aggregated to form embryoid bodies (EBs). We initiated mesoderm differentiation by addition of Wnt activator CHIR99021 (CHIR), as in monolayer differentiation protocols[2]. We defined the optimal time of CHIR addition based on EB diameter: EBs smaller than 100 μm fell apart upon CHIR incubation, and EBs bigger than 300 μm differentiated less efficiently (<90% TNNT2+; Fig. 1c; Supplementary Fig. S1c–e), likely due to inherent diffusion limits of larger EBs[4,26]. Therefore our protocol targets CHIR addition when EB diameter reaches 100 μm, which typically occurs at 24 hours.

We found that CHIR for 24 h followed by a gap of 24 h and then IWR-1 (IWR) for 48 h yielded optimal cardiac differentiation (Fig. 1a and Supplementary Fig. S1a). In 25 differentiations of 14 different hiPSC lines (6 lines from different donors and 8 gene-edited lines) treated with 7 μM CHIR and 5 μM IWR at these time points, we obtained on average ~1.21 million cells per mL (Fig. 1d) with >90% TNNT2+ cells (~2.4 hiPSC-CMs/input hiPSC; Fig. 1e). High percentages of TNNT2+ cells were further confirmed by analysis of cardiac markers TNNT2 and ACTN2 in cryosectioned bCMs at dd15 (Supplementary Fig. 1f; Supplementary Movie 1).

For functional comparison to bCMs, we used healthy control hiPSCs (WTC-Cas9; referred to as "control hiPSC-CMs") and differentiated them in parallel in adherent monolayers using the same protocol, except for a 48 h incubation period with CHIR instead of 24 h (Supplementary Fig. 1a′). Incubation with CHIR for 24 h led to failed monolayer differentiation. We did not apply hiPSC-CM enrichment methods[27] at the completion of either bCM or mCM differentiation protocols. Compared to bCMs, mCMs showed higher spontaneous beating frequency (Fig. 1f; Supplementary Movies 2 and 3), suggestive of lower maturity. Moreover, we obtained lower mCM yields (Fig. 1d) and higher intra- (Supplementary Movie 4) and inter-batch variability in cardiomyocyte purity (%TNNT2+ cells; Fig. 1e).

We first observed contraction in bCMs at differentiation day 5 (dd5; Supplementary Movie 5), versus dd7 in mCMs and previously reported suspension culture hiPSC-CM differentiations[1,4–6] (Supplementary Table 1). To validate this observation, we differentiated hiPSCs in which GFP is fused to endogenous sarcomere protein TNNI1[28] in the bioreactor and visualized onset of GFP expression

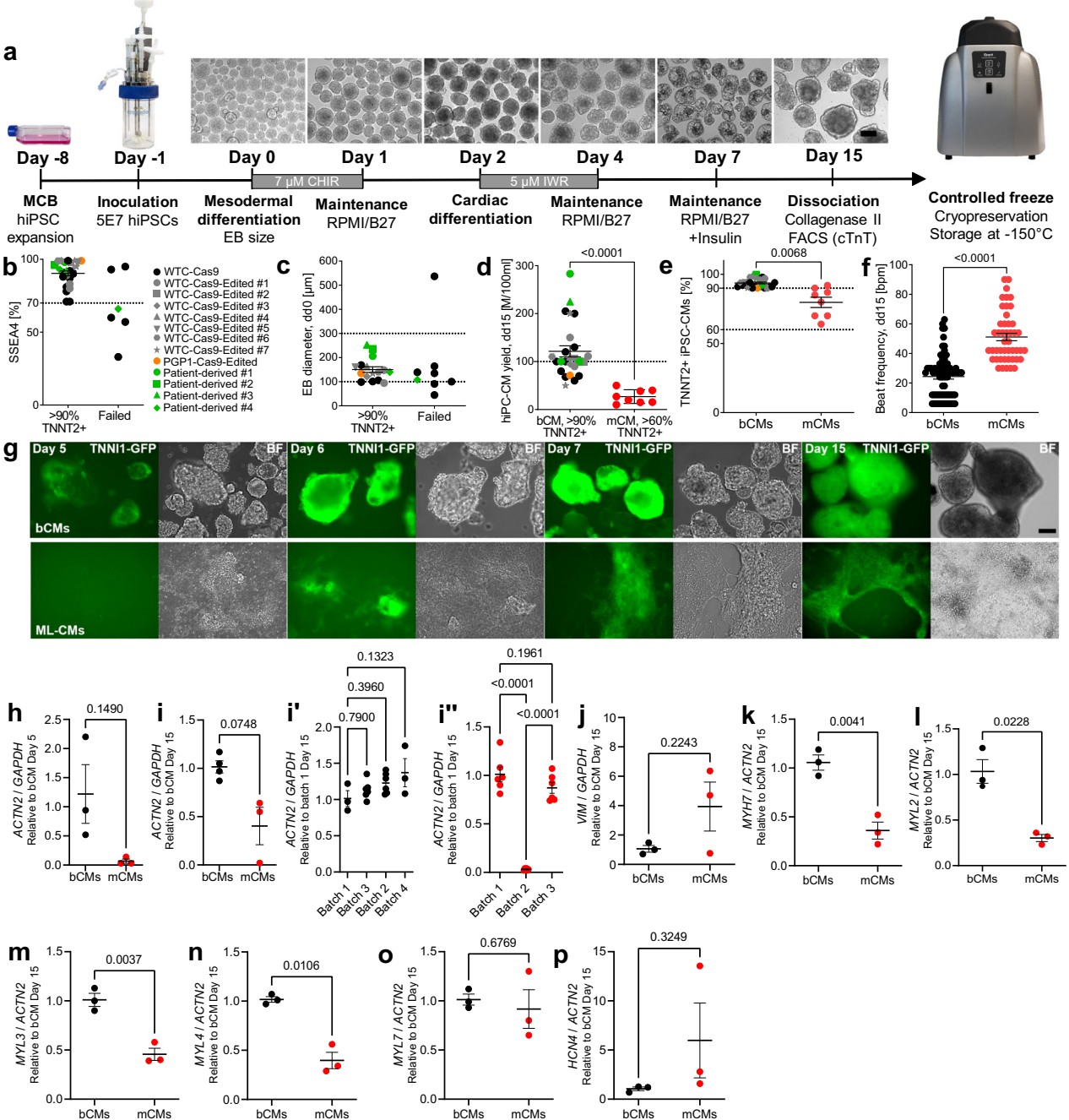

**Fig. 1 | Optimized stirred bioreactor cardiac differentiation protocol.**
**a** Schematic of the optimized bioreactor cardiac differentiation protocol.
**b**, **c** Characteristics of successful bioreactor differentiations. Runs were categorized as failed when they yielded <90% TNNT2+ hiPSC-CMs. Runs were categorized as failed when they yielded <90% TNNT2+ hiPSC-CMs. Cultures with low frequency of SSEA4 by flow cytometry (**b**; n = 31 differentiations) or out of range mean EB diameter (**c**; n = 24 differentiations) had higher failure likelihood. **d**, **e** hiPSC-CM yield (**d**) and purity (**e**) at dd15 (bioreactor, n = 25 differentiations; monolayer, n = 8 differentiations). Percentage of cells positive for cardiomyocyte marker TNNT2 was measured by flow cytometry. **f** Spontaneous beating frequency of bCMs and mCMs at dd15 (n=number of differentiations; number of EBs or wells: bCM (n = 3; 71); mCM (n = 3; 46). **g** Timeline of bioreactor and monolayer cardiac differentiation monitored using TNNI1-GFP hiPSCs (bar: 200 µm). **h–p**. RT-qPCR analysis of marker gene expression during bioreactor and monolayer cardiac differentiation. Marker genes were: *ACTN2*, cardiomyocytes; *VIM*, non-cardiomyocytes; *MYH7, MYL2* and *MYL3*, ventricular cardiomyocytes; *MYL4* and *MYL7*, atrial cardiomyocytes; *HCN4*, developmentally repressed pacemaker channel. Expression is relative to dd5 bCM (**h**; n=number of differentiations: bCMs, n = 3; mCMs, n = 3) or dd15 bCM (**i–p**: bCMs, n = 3–4; mCMs, n = 3). Data are expressed as mean ± SEM. Points represent biological replicates for each independent differentiation, except for **i'–i''** in which points represent technical replicates from separate batches. **d–f**, **h**, **i**, **j–p**: two-tailed Welch's unpaired *t* test. **i'–i''**: one-way ANOVA with Tukey´s post-test. MCB, Master cell bank; hiPSC, human induced pluripotent stem cells; CM, cardiomyocyte; BF, bright field.

(Fig. 1g). We first observed GFP in bCMs on dd5, in contracting areas at the edges of EBs. mCMs first showed GFP expression at day 6 (Fig. 1g), but these GFP⁺ areas did not visibly contract until dd 7. By RT-qPCR, *ACTN2* was expressed in dd5 bCMs, whereas its level in mCMs was far lower (Fig. 1h). Additionally, bCMs had less inter-batch variation in *ACTN2* levels compared to mCMs at day 15 (Fig. 1i–i''). Mesenchymal marker vimentin (*VIM*) was lower in dd15 bCMs, although the difference did not reach significance due to variation in mCMs (Fig. 1j).

Furthermore, we found significantly higher expression of ventricular markers *MYH7*, *MYL2* and *MYL3* (Fig. 1k–m) and a fraction of 83.4% bCMs stained for ventricular myosin light chain (MLC2v) using flow cytometry (Supplementary Fig. 1g). Atrial markers *MYL4* and *MYL7* were higher or unchanged, respectively, in bCMs compared to mCMs (Fig. 1n, o). Finally, *HCN4*, encoding a developmentally repressed pacemaker channel, was higher in some mCM batches with high interbatch variability (Fig. 1p). Finally, TNNT2 protein levels were significantly higher in dd15 bCMs compared to mCMs (Supplementary Fig. 1h).

The ability to cryopreserve and recover viable cells that retain functional properties is critical to incorporate large scale differentiation protocols into efficient workflows. We optimized freeze and thaw protocols by adjusting cell dissociation protocols, cryo-protectant media, freezing conditions, and thawing procedures, and used a dedicated computer-controlled cell freezer (Supplementary Table 5; see Methods). Our freeze/thaw protocol yielded ~94% viable cells (trypan blue negative) after cryo-recovery, with plating efficiencies of ~51% for bCMs and ~46% for mCMs (Supplementary Fig. 1i, j). Functional testing of cryo-recovered hiPSC-CMs is discussed in subsequent sections.

Together, we established an integrated bioreactor-based workflow that yields consistently high numbers of highly pure hiPSC-CMs and developed methods for efficient cryo-preservation and cryo-recovery.

## Cell composition of bCMs and mCMs

To gain a better understanding of generated cell types and differences between bCMs and mCMs, we performed single cell RNA sequencing (scRNAseq) of freshly dissociated hiPSC-CMs at dd15. Using microdroplet technology, we captured single cell transcriptomes of bCMs and mCMs, each in biological duplicate. From a total of 5173 bCM and 2513 mCM high quality cell transcriptomes, unsupervised clustering on the most variable genes revealed 11 cell clusters and excellent agreement between biological duplicates (Fig. 2a and Supplementary Fig. 2a, b; Supplementary Data 1). Based on expression of canonical marker genes, the clusters were identified as cardiomyocytes (Clusters: 0, 1, 2, 4, 5, 8, 9), skeletal muscle cells (Cluster: 6), smooth muscle cells (Cluster: 76), non-cardiomyocytes (Cluster: 3), and endothelial cells (Cluster: 10). The cardiomyocyte fraction was markedly higher in bioreactor (88%) compared to monolayer (51%) differentiation (Fig. 2a, b and Supplementary Fig. 2b). We used canonical marker genes to classify the cardiomyocyte clusters. Clusters 0 and 1, highly enriched for ventricular marker genes *MYH7*, *MYL2*, and *MYL3* were enriched in bioreactor (67% of cardiomyocytes) compared to monolayer differentiation (44% of cardiomyocytes; Fig. 2c; Supplementary Fig. 2b). Conversely, cluster 2, containing cardiomyocytes with high expression of atrial marker genes *MYL4* and *MYL7*, were less frequent in bioreactor (8% of cardiomyocytes) compared to monolayer differentiation (36% of cardiomyocytes). Assignment of ventricular and atrial types was further supported by calculating chamber type scores that aggregated the expression of multiple marker genes (Fig. 2d). Comparison of the genes differentially expressed between bCM and mCM cells within the principal cardiomyocyte clusters (0, 1, 2; Fig. 2e) showed that genes more highly expressed in bCMs were strongly enriched for electron transport chain, mitochondrial respiratory chain complexes, and muscle contraction (Fig. 2f, left). In contrast, genes more highly expressed in mCMs were highly enriched for glycolysis, extracellular matrix organization, and heart development (Fig. 2f, right). We validated that bCMs expressed higher levels of mitochondrial metabolism genes *HADHA* and *ACADVL* than mCMs (Supplementary Fig. 2c, d), consistent with an upregulation of these genes in a prior report on suspension culture[19]. Upregulation of these genes was previously associated with a maturation protocol based on the induction of PPARdelta in hiPSC-CMs, which improved functional output[29].

The non-cardiomyocyte fraction was dramatically lower in bioreactor compared to ML differentiation (12% vs. 49%; Fig. 2b, b'). An endothelial cell population marked by *PECAM1*, *CDH5* and *HLX* was uniquely found in bioreactor differentiation (Fig. 2c; Supplementary Fig. 2b). mCMs were highly enriched for non-cardiomyocyte (non-CM) cluster 3 (bCMs: 3% vs mCMs: 20%), which mainly expressed fibroblast marker genes (*COL3A1*, *COL1A1*, *FN1*). (Fig. 2a, Supplementary Fig. 2b, Supplementary Data 1).

Taken together, scRNAseq analysis of cellular composition indicated that bioreactor differentiation yields a higher fraction of hiPSC-CMs, and these hiPSC-CMs have more mature gene expression profiles and greater ventricular identity, consistent with the RT-qPCR analysis (Fig. 1h–p). Additionally, all non-CM clusters showed robust marker expression in bCMs compared to mCMs, indicating higher cellular non-CM specification in bCMs (Supplementary Fig. 2b).

## Functional characterization of bCMs in 2D assays

For morphological and functional analysis, cryopreserved control hiPSC-CMs were thawed and plated in 96-well plates pre-coated with diluted Geltrex (see Methods). After 7 days, unpatterned hiPSC-CMs were fixed and morphologically analyzed. Staining for ACTN2 showed that many bCMs had elongated morphology, reminiscent of the rod shape of mature, *de facto* human cardiomyocytes (Fig. 3a). Quantification of bCM circularity confirmed their consistent elongated morphology across multiple batches. In comparison, mCMs displayed greater circularity and higher inter-batch morphological variation (Fig. 3b; Supplementary Fig. 3a). bCM cell area was not significantly different than mCMs (bCMs: 1752 ± 112.3 μm$^2$; mCMs: 1905 ± 246.6 μm; Supplementary Fig. 3b), and inter-batch variation in cell area was significantly less than mCMs (Fig. 3c). Measured cell areas were comparable to other hiPSC-CM control lines cultured for 7[18] and 30 days[15,17]. Mature, *de facto* human cardiomyocytes are 80% mononucleated[30], and multinucleation tends to increase with cardiac disease[10,17,31]. Accordingly, both bCMs and mCMs were predominantly mononuclear. However, mCMs had elevated frequency of binucleated or multinucleated cardiac cells (Fig. 3d). Moreover, a greater fraction of mCM nuclei exhibited H2AFx immunoreactivity, a marker of DNA double strand breaks (Fig. 3e).

Plating hiPSC-CMs onto contact printed rectangular extracellular matrix (ECM) islands promotes their structural maturation, including alignment of sarcomeres perpendicular to the cell's long axis[32]. We compared cryo-recovered bCMs to mCMs after seeding onto rectangular ECM islands with the 7:1 aspect ratio of mature adult human cardiomyocytes. Cells were fixed on day 1, 3 and 7 after plating and stained for cardiac marker ACTN2 (Fig. 3f and Supplementary Fig. 3c). Cell attachment and sarcomere alignment were quantified by unbiased computational image analysis[33]. bCMs better survived plating on the micropatterned substrates, as demonstrated by their markedly higher coverage at all timepoints compared to mCMs (Fig. 3g; Supplementary Fig. 3d). Sarcomere packing density and orientation order parameter, two different measures of sarcomere alignment[33], were considerably higher in bCMs than in mCMs at all investigated timepoints (Fig. 3h and Supplementary Fig. 3e, f). Compared to unpatterned cells (Fig. 3d), a greater proportion of patterned bCMs and mCMs were bi- and multinucleated (Fig. 3i). Staining for DNA double strand break marker H2AFx indicated strikingly higher levels in patterned mCMs compared to patterned bCMs (Fig. 3j, k) or to unpatterned cells (Fig. 3e). Additionally, unbiased analysis of nuclear morphology[34] identified a significantly higher fraction of nuclei with abnormal morphology in mCMs for all investigated timepoints (Supplementary Fig. 4a–g). These data indicate that micropatterned substrates increased morphological maturation of bCMs, in line with previous findings on fresh mCMs[35,36]. bCMs were more amendable to single cell micropatterning

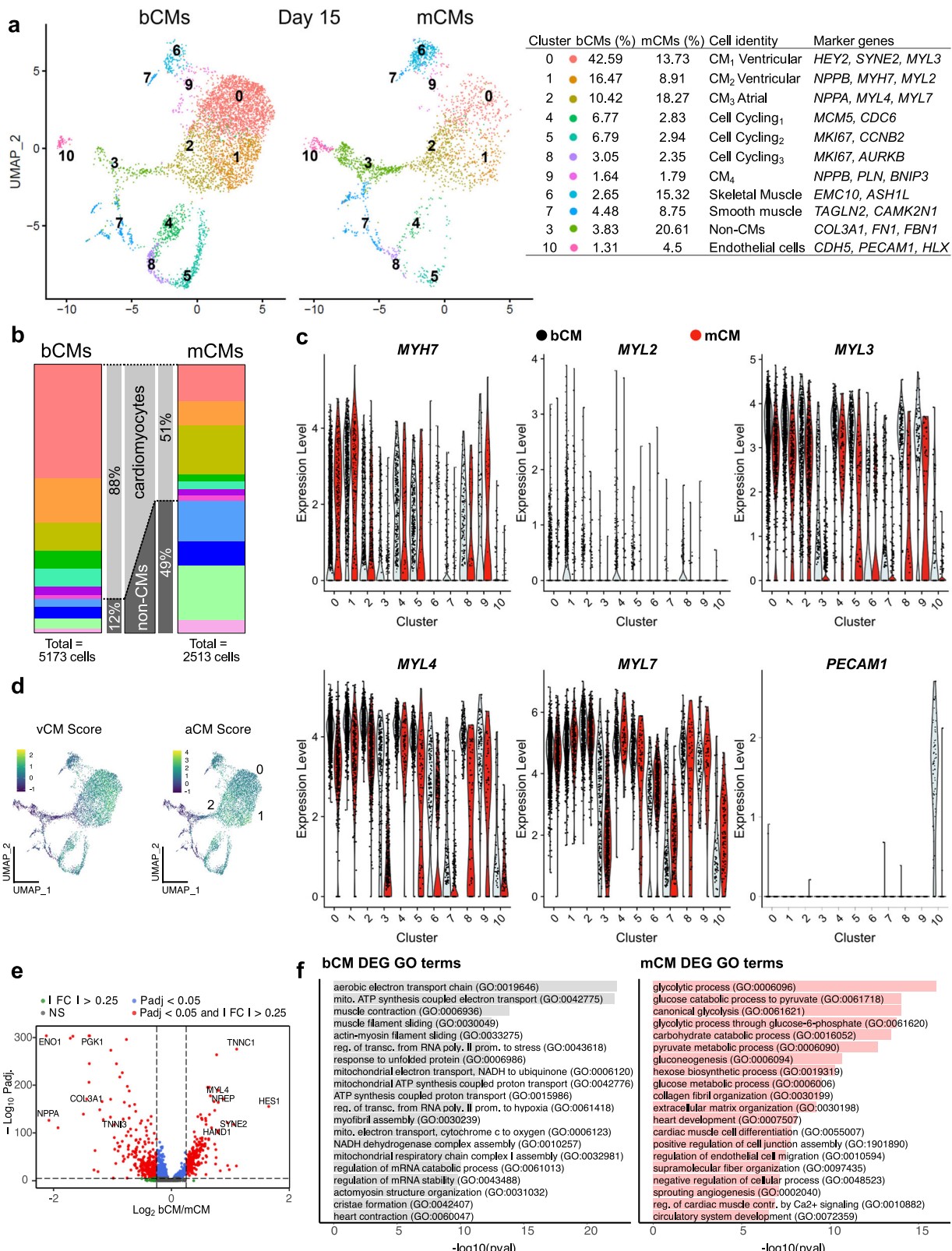

than mCMs, with greater survival and sarcomere assembly, and reduced manifestations of genotoxic stress.

Next, we analyzed the physiological properties of bCMs compared to mCMs. We recorded $Ca^{2+}$ transients by loading hiPSC-CMs with the $Ca^{2+}$ sensitive dye Fluo-4 and electrically pacing them for 10 seconds. mCMs failed to follow electrical pacing (1 Hz; n = 4 batches; Supplementary Fig. 5a) and were therefore excluded from the

analysis. In contrast, bCMs were reliably captured by the same pacing protocol. $Ca^{2+}$ transients showed high inter-batch reproducibility and consistent responses to beta-adrenergic stimulation by iso-proterenol (Iso) (Supplementary Fig. 5b–f). Next, we recorded bCM action potentials (APs) using Fluovolt, a membrane voltage sensitive dye, and pacing at 1 Hz. bCMs displayed typical ventricular AP morphology (Supplementary Fig. 5g), corroborating our RT-qPCR

**Fig. 2 | scRNAseq reveals higher cardiomyocyte content and degree of cellular specification in bCMs. a** scRNA-seq UMAP clustering of mCM and bCM cultures showing 11 clusters, marker genes, assigned cell types, and distribution in bCM and mCM cultures. **b** Stacked bar graph showing cellular composition of mCM (right) and bCM (left) cultures. Clusters are divided into cardiomyocytes (top) and non-cardiomyocytes (bottom). Color coding is the same as in **a**. **c** Violin-plots showing the relative expression of a subset of cardiac and non-cardiac marker genes (y-axis) across all clusters for bCMs (grey) and mCMs (red). **d** Composite ventricular and atrial cardiomyocyte (vCM, aCM) scores derived from multiple marker genes. Clusters 0 and 1 and greater vCM score, and cluster 2 had greater aCM score. **e** Differentially expressed genes (DEGs) between cardiomyocytes in bCM and mCM cultures. Significances were calculated using Wilcoxon rank-sum test (implemented by the Seurat findmarkers function), with p values adjusted using the Benjamini-Hochberg procedure (for DEGs, FDR < 0.05). **f** Top biological process gene ontology (GO) terms for DEGs more highly expressed in bCMs (left) or mCMs (right).

(Fig. 1o, p) and scRNAseq findings (Fig. 2). As expected, adrenergic stimulation with Iso shortened action potential duration (Supplementary Fig. 5h).

Although mCMs failed to follow electrical pacing 4 days after cryo-recovery, we found that they did after 7 days and therefore repeated the $Ca^{2+}$ transient and action potential measurements at this time point. Lactate treatment has been established as a method to enrich hiPSC-CMs and enhance their metabolic maturation[37]. Therefore we compared bCMs to mCMs without (-L) and with lactate treatment (+L), each with freezing and cryo-recovery ("cryo") or without ("fresh"). mCMs included in these studies were already ~80% $TNNT2^+$ prior to lactate treatment, which was slightly increased by lactate (Supplementary Fig. 6a). However, lactate did enhance mCM functional properties (see below). Immunostaining showed that bCMs formed a confluent monolayer of cardiomyocytes, whereas mCMs with or without lactate treatment formed cardiomyocyte patches surrounded by mesenchymal cells (Supplementary Fig. 6b).

We compared $Ca^{2+}$ transients between fresh or cryo bCMs, mCMs-L, and mCMs+L (Fig. 3l–n and Supplementary Fig. 6c–g). Cryo bCMs had $Ca^{2+}$ transients with the highest upstroke and recovery velocities, longest duration, and greatest amplitude. In comparison, cryo mCMs-L or mCMs+L had lower upstroke and recovery velocities, duration, and amplitude. Comparison of fresh to cryo showed different effects on bCMs compared to mCMs: $Ca^{2+}$ transients in cryo bCMs had higher upstroke velocity, decay time, duration and amplitude than fresh bCMs, whereas $Ca^{2+}$ transients in cryo mCMs+L had lower decay time, and duration than fresh mCMs+L. Cryo and fresh mCMs were comparable. The mCMs+L result was in keeping with a prior report of unchanged $Ca^{2+}$ transient amplitude after cryo-recovery in some hiPSCs differentiated in monolayer conditions with lactate treatment, and contrasts with our finding that cryo-recovery increased bCM $Ca^{2+}$ transient amplitude.

We similarly compared action potentials between these groups (Fig. 3o–q and Supplementary Fig. 6h–l). Cryo bCMs had higher action potential upstroke velocity and recovery velocity compared to cryo mCMs±L. Action potential upstroke velocity was also higher in cryo compared to fresh bCMs. Cryo and fresh mCMs were comparable. Unlike $Ca^{2+}$ transient parameters, action potential parameters tended to be less affected by cryo-recovery across bCMs, mCMs+L, and mCMs-L.

To assess mitochondrial function, we measured cellular oxygen consumption rate (OCR) during the sequential addition of mitochondrial inhibitors (mitochondrial stress test). Cryo-recovered and freshly plated bCMs had higher basal respiratory OCR, maximal respiratory OCR, and ATP production than mCMs-L (Fig. 3r and Supplementary Fig. 6m–o). These observations are consistent with the higher level of mitochondrial metabolism genes *HADHA* and *ACADVL* in bCMs (Supplementary Fig. 2d, e). Lactate treatment significantly increased mCM mitochondrial function to a level comparable to bCMs (Fig. 3r and Supplementary Fig. 6m–o). These data indicate that bCMs intrinsically had metabolic maturation comparable to mCMs after lactate treatment.

Taken together, these functional data indicate that cryo bCMs have robust physiological cardiomyocyte properties.

## Functional characterization of bCMs in EHTs

3D culture of hiPSC-CMs in fibrinogen gels subjected to anisotropic stress promotes cardiomyocyte maturation and sarcomere organization[5,15]. These engineered heart tissues (EHTs) also facilitate the measurement of cardiomyocyte force development and relaxation. We assembled EHTs using cryopreserved control bCMs (Supplementary Movie 6) and mCMs without lactate treatment (Supplementary Movie 7; Fig. 4a). From days 5 to 32 after EHT casting, we recorded EHTs during spontaneous beating. bCM and mCM EHTs had similar spontaneous beating frequencies (Fig. 4b). bCM EHTs generated greater force than mCM EHTs at all investigated timepoints (Fig. 4c) and considerably exceeded previously reported values using the same EHT constructs. In contrast, force generated by mCM EHTs was comparable to prior values[5,15]. Compared to mCM EHTs, bCM EHTs converged on similar contraction kinetics, as measured by the 50% contraction time (C50, Fig. 4d), and greater relaxation rates, as measured by the 50% and 90% relaxation time (R90, Fig. 4e and R50, Supplementary Fig. 7a).

We recorded EHTs during optogenetic pacing at 1–3 Hz (Supplementary Movie 8). At any given pacing frequency, bCM EHTs were captured more frequently than mCM EHTs (Supplementary Fig. 7b). Under all pacing conditions, force was higher in bCMs EHTs compared to mCM EHTs (Supplementary Fig. 7c), and C50 did not significantly differ between these groups (Supplementary Fig. 7d). R90 was significantly lower in bCM EHTs at baseline and 2 Hz pacing; at 1 Hz and 3 Hz, bCM EHTs likewise had lower R90 values, although statistical significance was difficult to evaluate due to the low number of mCM EHTs captured at these rates (Supplementary Fig. 7e). An independent set of EHTs was similarly analyzed in Tyrode solution, with similar findings (Fig. 4f–h; Supplementary Fig. 7f). Under these conditions, both bCM and mCM EHTs generated greater force compared to culture medium, most likely due to higher calcium concentrations in Tyrode solution (bCMs: 0.374 mN vs 0.537 mN, mCMs: 0.104 mN vs 0.177 mN). Furthermore, in Tyrode solution a subset of bCM EHTs was successfully captured at 4 Hz pacing (Supplementary Fig. 7g–j; Supplementary Movie 9), notably faster than the previously reported maximal rates achieved for EHTs[38–40] without physical conditioning[41,42]. Given greater force generation by bCMs in EHTs, we stained EHT cross-sections for cardiac sarcomere proteins ACTN2 and TNNT2. Consistent with the physiological data, we found greater sarcomere density in bCM compared to mCM EHTs (Fig. 4i). Furthermore, sarcomere length was higher in bCMs compared to mCMs EHTs (1.75 vs 1.64 μm; Fig. 4j), indicating greater maturity of bCMs sarcomeres.

Since lactate treatment enhanced physiological properties and metabolic maturation of mCMs, we also generated mCM+L EHTs and compared them to bCM and mCM-L EHTs. We made serial measurements at days 5, 20, 22, and 32. EHT spontaneous beating frequencies were higher in mCMs+L than mCMs-L or bCMs (Supplementary Fig. 8a). mCM+L and mCM-L EHTs generated comparable force, which was much lower than bCM EHTs after day 20 (Supplementary Fig. 8b). mCM+L EHTs showed shorter contraction time than bCM EHTs, likely reflecting markedly elevated bCM developed force, and comparable relaxation kinetics (Supplementary Fig. 8c, d).

Overall, greater function of bCMs EHTs was observed when compared to mCM EHTs with and without lactate treatment.

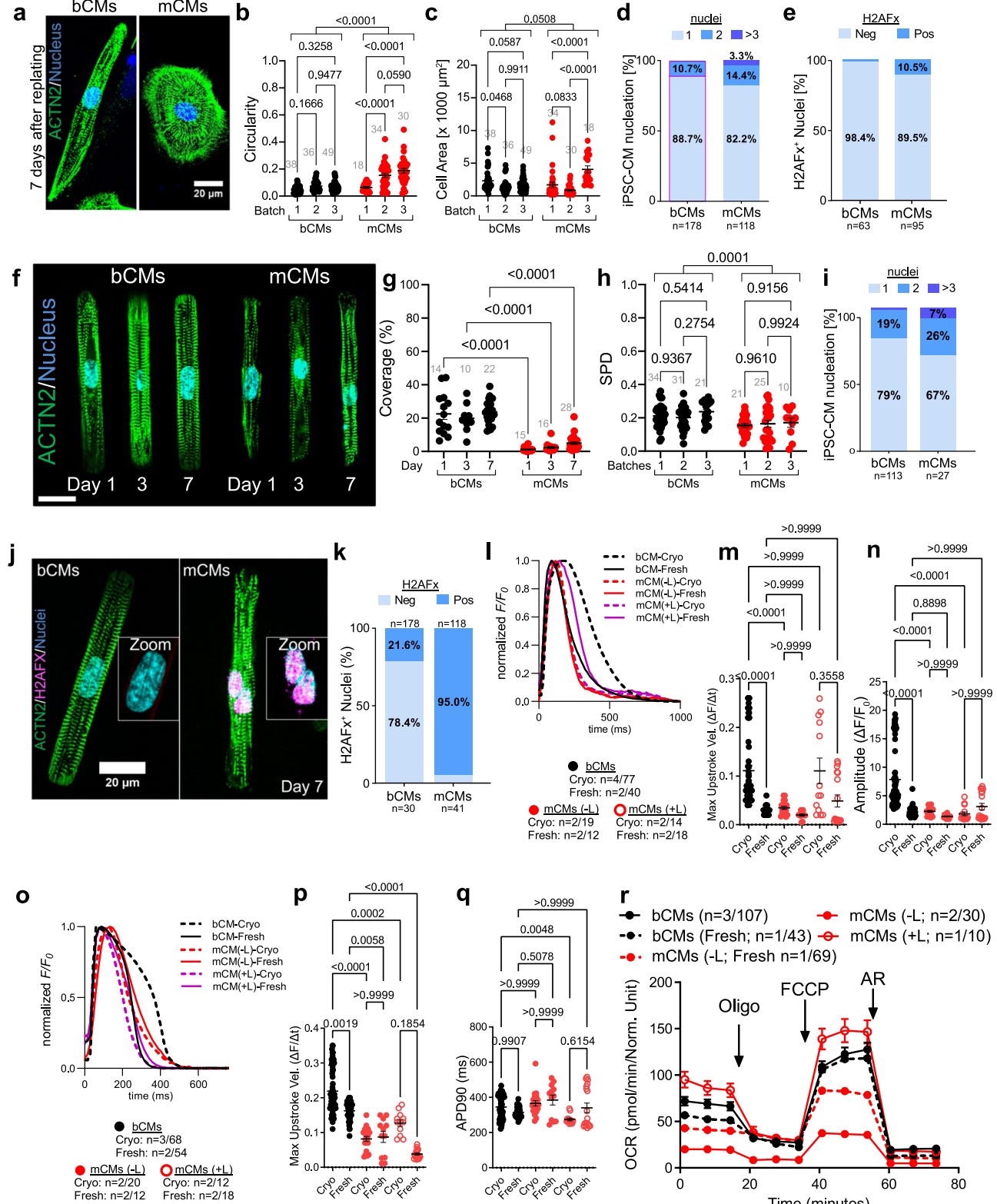

## Suspension culture differentiation in spinner flasks

Compared to bioreactors, magnetically stirred suspension spinner flasks are more widely available and offer greater culture volume flexibility. However, spinner flasks are not capable of real time monitoring and adjustment of pH, oxygen, temperature, and $CO_2$. We found that the suspension culture protocol performed well in spinner flasks with minor adaptations (Fig. 5a). hiPSCs subjected to our optimized workflow showed high SSEA4 values (>70%, Fig. 5b). In contrast

to the bioreactor, optimal EB size ranged from 80–100 μm for CHIR addition (Fig. 5c). In six differentiations of two different hiPSC lines, differentiation with 7 μM CHIR and 5 μM IWR yielded on average 1.8 million cells/ml at dd15, although yield varied more for spinner flask-differentiated cardiomyocytes (sCMs) than bCMs (Fig. 5d). Cardiomyocyte purity of sCMs (94 ± 1.8% TNNT2+ cells; 3.58 hiPSC-CMs/input hiPSC; Fig. 5e) was comparable to bCMs. However, spontaneous beating frequency was higher for sCMs compared to bCMs (Fig. 5f).

**Fig. 3 | Comparison of bCMs and mCMs on 2D platforms. a–e** Morphological characteristics of unpatterned hiPSC-CMs. Cryo-recovered bCMs and mCMs were cultured for 7 days on unpatterned Geltrex-coated dishes and then stained for sarcomere Z-line marker ACTN2. **a** Representative images illustrate elongated shape of bCMs compared to mCMs. Bar, 20 μm. **b, c** Circularity and cell area were quantified from 3 independent differentiation batches of bCMs and mCMs. Grey numbers indicate cells analyzed. Two-way ANOVA with Tukey's post-test. **d** Nucleation of unpatterned bCMs and mCMs after 7 days in culture. Chi-squared p < 0.0001. **e** Unpatterned bCMs and mCMs stained for H2AFX, a marker of DNA damage response. Chi-squared *p* < 0.0001. **f–k** Cryo-recovered cells were plated on extracellular matrix rectangular islands. After 1, 3, and 7 days, samples were fixed and stained. **f** Representative images. Bar, 20 μm. **g** Quantification of single cell islands covered by bCMs or mCMs. Grey numbers indicate 10x fields analyzed. One-way ANOVA with Šidák's post-test. **h** Sarcomere organization of micropatterned bCMs and mCMs measured using sarcomere packing density after 7 days on micropatterned substrates. Grey numbers indicate cells analyzed. Two-way ANOVA with Šidák´s post-test. **i** Nucleation of micropatterned bCMs and mCMs after 7 days in culture. Chi-squared p < 0.0001. **j, k** DNA damage response in micropatterned bCMs and mCMs. **j** Representative images. Bar, 20 μm. **k** Quantification of H2AFX staining in bCMs (n = 30) and mCMs (n = 41). Chi-squared p < 0.0001. **l–n** $Ca^{2+}$ transients were recorded under 1 Hz electrical pacing 7 days after plating. **l** Average, normalized $Ca^{2+}$ transients. **m** Maximum upstroke velocity. **n** $Ca^{2+}$ transient amplitude. Kruskal–Wallis with Dunn's multiple comparison test. **o** Average, normalized action potentials. **p** Maximum upstroke velocity. **q** action potential duration at 90% recovery (APD90). Kruskal–Wallis with Dunn's multiple comparison test. **r** Mitochondrial stress test. Cells were cultured in 96 well dishes designed to measure oxygen consumption rate (OCR). Arrows indicate addition of oligomycin, carbonyl cyanide-4 (trifluoromethoxy) phenylhydrazone (FCCP), and antimycin/rotenone (AR). Sample sizes indicate number of differentiations/number of cells (**b, c, g, h**) or replicates (**l–r**). Data are expressed as mean ± SEM. Source data are provided as a Source Data file.

Taking advantage of the culture volume flexibility of spinner flasks, we explored the ability of the optimized suspension culture protocol to scale to larger volumes. Scaling from our standard culture volume of 100 ml by 2.5 (125 million hiPSC in 250 mL volume) or 3.8 fold (190 million hiPSC in 380 mL volume) resulted in greater than linear increases in yield: 250 ml yielded 600 million (2.4 million/ml; 2.9 sCMs/input hiPSCc) and 380 ml yielded 1320 million (3.4million/ml; 6.9 sCMs/input hiPSC) (Fig. 5d, e).

As with bioreactor differentiation, spontaneous beating was observed at dd5. Likewise, *ACTN2* expression was expressed at comparable levels between sCMs and bCMs at dd5 (Fig. 5g). At dd15, sCMs expressed greater *ACTN2* but with greater inter-batch variability (Fig. 5h–h'). Mesenchymal marker *VIM* was significantly lower in dd15 bCMs (Supplementary Fig. 9a). Ventricular markers *MYH7* and *MYL2* (Fig. 5i, j) were decreased in sCMs when compared to bCMs, although the *MYH7* difference was not significant due to sCM variability. Atrial markers *MYL4* and *MYL7* were significantly lower or unchanged, respectively, in sCMs (Fig. 5l, m). The pacemaker gene *HCN4* was not significantly different between bCMs and sCMs (Fig. 5n).

These data indicate that spinner flasks are an economical alternative to bioreactors and enable flexible culture volume scaling.

## Comparison of sCMs and bCMs functional properties

For functional analysis of sCMs, we repeated $Ca^{2+}$ transient and APD measurements with fresh and cryo-recovered control sCMs after 7 days of culture. Cryo-recovered sCMs were highly viable (-94.6%; Supplementary Fig. 9b) and formed confluent CM monolayers (Supplementary Fig. 9c), comparable to bCMs for both conditions (Supplementary Fig. 6b). In sCMs, $Ca^{2+}$ transients and APDs had high upstroke velocities and $Ca^{2+}$ transient amplitudes (Fig. 5o–q, S9d-h and Fig. 5r–t, Supplementary Fig. 9i–m) that were more similar to bCMs than mCMs. Consistent with the bCM results, $Ca^{2+}$ transient amplitude was greater in cryo compared to fresh sCMs (Fig. 5q). However, no changes were found between cryo and fresh sCMs in upstroke velocities (Fig. 5p). Notably, we found higher reproducibility between cryo and fresh sCMs when compared to bCMs.

To compare mitochondrial function of sCMs to bCMs, we performed mitochondrial stress tests. These assays showed no significant differences between sCMs and bCMs (Supplementary Fig. 9n–q).

Next we compared the performance of EHTs assembled from cryo sCMs and bCMs (Fig. 5u). During continuous culture for 32 days, sCM EHTs generated less force than bCM EHTs (sCM 0.158 mN vs bCM 0.374 mN) but more than mCM EHTs (0.104 mN Supplementary Fig. 9r). EHT spontaneous frequency became lower for sCMs than bCMs at day 32. Times for 50% contraction (C50) or 90% relaxation (R90) were comparable (Fig. 5v–y).

These data show that sCMs share many of the functional characteristics of bCMs, although some parameters, particularly EHT force, were superior for bCMs.

## Generation of bioreactor-derived cardiac organoids (bCOs)

By minor adjustment of our suspension culture differentiation protocol, we observed the emergence of larger spheroids that resembled previously published self-organized cardioids in size and morphology[20,21]. We named these spheroids "bioreactor-derived cardiac organoids" (bCOs; Supplementary Movie 10). Approximately 10% of all EBs formed at day 0–2 developed into spherical bCOs under these adjusted culture conditions (Supplementary Fig. 10a). The remaining EBs formed aspherical structures or aggregated during prolonged culture and were not analyzed further (Supplementary Fig. 10a). Assessment of spherical bCOs cultured for 15 days (Fig. 6a) revealed an average diameter of 1.8 ± 0.8 mm (Fig. 6b), spontaneous beating (16.3 ± 4.8 beats per minute; Fig. 6c), and 2-fold higher *ACTN2* mRNA expression when compared to dd15 mCMs (Fig. 6d). bCOs formed wall-lined cavities divided by septae, reminiscent of cardiac chambers (Fig. 6e; Supplementary Fig. 10b Supplementary Movie 11). Most cells in bCO walls and septea expressed TNNT2 (Fig. 6f; Supplementary Fig. 10c). Closer analysis of the bCO walls showed sarcomere striation with anisotropic alignment (Supplementary Fig. 10d).

Morphological observation of bCOs indicated that most bCO spheroids initially remodel into biconcave discs (S10e,f) and then form "doughnut-shaped" bCOs by day 2 (S10g). This hole was then filled in, ultimately yielding a central chamber divided by a small number of septae that persisted throughout the culture duration of bCOs (Supplementary Fig. 10h), which could be extended to over 120 days.

We used scRNAseq to further interrogate the cellular composition and transcriptomic states of bCOs. 4032 high-quality single cell transcriptomes were integrated with bCM and mCM scRNAseq data. Similar to bCM cultures (Fig. 2b, c), 88% of cells were assigned to 6 cardiomyocyte clusters (0, 1, 2, 3, 6, 7), with the majority of cells in the ventricular cardiomyocyte clusters (0, 1), and 12% to non-CM clusters (4, 5, 8, 9, 10; Fig. 6g, h). Immunofluorescence staining of cryosectioned bCOs confirmed fibroblast (cluster 4: THY1⁺), endothelial (cluster 9: PECAM1⁺), and non-CM vimentin (VIM⁺) expressing cells (cluster 4, 5, 8, 9) at day 15 (Supplementary Fig. 10i). However, these non-CMs were very infrequent, in line with the scRNAseq results (Fig. 6h). Therefore, bCOs contain predominantly cardiomyocytes and a small fraction of non-CMs, comparable to previously published cardiac organoids generated in static culture without any further modification of basic cardiac differentiation media[20,21,43]. Taken together, bCOs represent the first self-organizing human heart organoid completely generated in suspension culture.

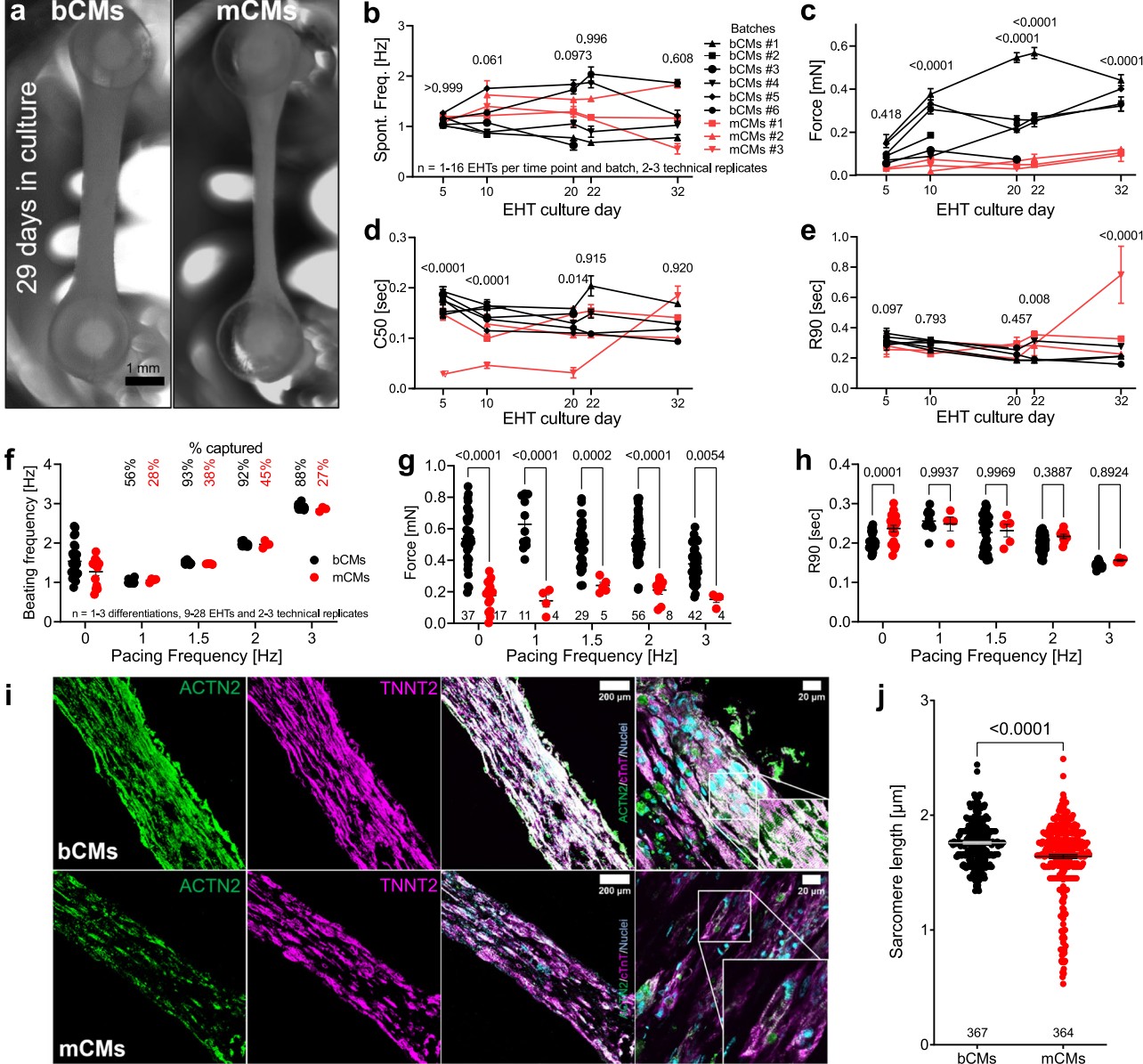

**Fig. 4 | Analysis of bCM and mCM function in 3D engineered heart tissues (EHTs). a** Representative images of EHTs after 29 days in culture (Scale bar, 1 mm). **b–e** Spontaneously beating EHTs assembled from cryo-recovered bCMs or mCMs were recorded in culture medium at 37 °C 5 to 32 days after EHT fabrication. **b** Spontaneous beating frequency. **c** EHT force. **d** Time to 50% contraction (C50). **e** Time to 90% relaxation (R90). Two-way ANOVA with Šidák´s post-test at each timepoint. **f–h** Analysis of EHTs in Tyrode solution without pacing (0 Hz) or with 1–3 Hz pacing. **f** EHT beat frequency in response to pacing. Only EHTs captured by pacing are shown. The percent of EHTs captured at each pacing rate is indicated. Two-way ANOVA with Šidák´s post-test. **g** Paced EHT force. **h** Paced EHT R90. **i** Histological characterization of bCM and mCM EHTs. After 34 days, EHT cryosections were stained for sarcomere Z-line protein ACTN2 and cardiac troponin T (TNNT2). Representative cryosections showed higher cellularity and greater sarcomere content and organization in bCM EHTs. **j** Sarcomere length. Quantification from 66 (bCM) or 59 (mCM) regions of interest in 4 (bCM) or 5 (mCM) EHTs from two independent differentiations. EHTs fixed at days 32, 34, and 39 were investigated and pooled for this analysis. Number of sarcomere intervals measured are indicated in the graph. Two-tailed Welch's unpaired *t*-test. Data are expressed as mean ± SEM. Source data are provided as a Source Data file.

## Discussion

Although cardiac differentiation protocols have been continuously improved over the past decade, current commonly used protocols suffer from limited scalability, high cost, inter-batch and even inter-well variation in efficiency and structural and functional properties, and some report loss of key functional properties upon recovery from cryopreservation[1,3–7,44] (Supplementary Table 1). These issues have presented substantial practical hurdles to performing reproducible and rigorous research using hiPSC-CMs. Labs relying on these cells invest considerable resources in their continuous culture and differentiation and accounting for well and batch effects. Prior suspension culture and bioreactor protocols offered a potential solution with improved reproducibility and scalability, yet lacked robust methods for cryo-recovery, and functional properties of the resulting cells were uncharacterized. Here we present a relatively low-cost bioreactor-based cardiac differentiation protocol (~$569 per bioreactor run yielding ~121 million hiPSC-CMs, compared to ~$1075 for ML differentiation with comparable yield, exclusive of manpower; Supplementary Table 2) with defined benchmarks and controlled freeze/thaw procedures. We extensively characterized the resulting bCMs and demonstrated that they have robust transcriptional, structural, metabolic and functional properties of ventricular cardiomyocytes,

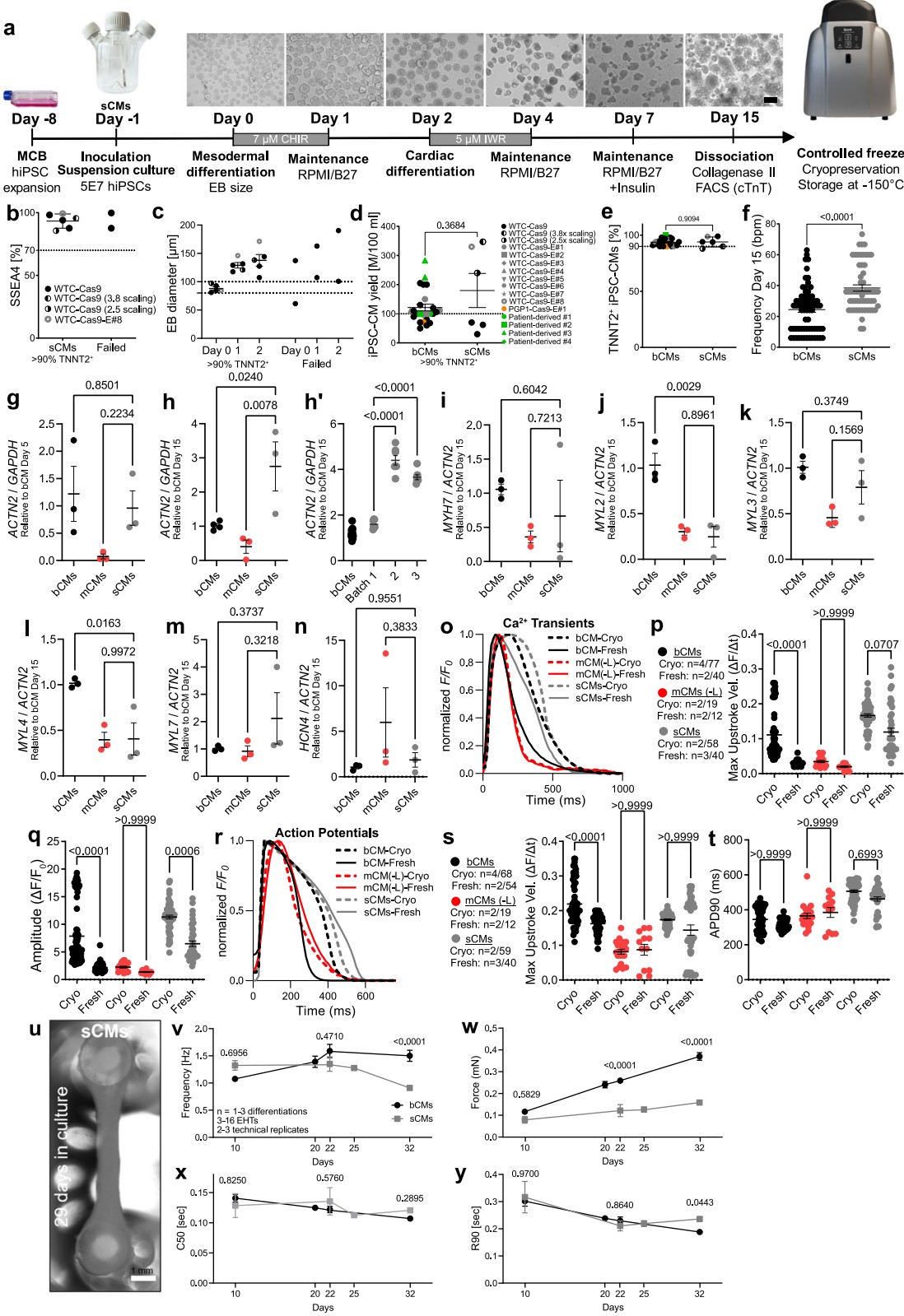

reduced batch-to-batch variation, and greater force production in EHTs than previously described[5,15–17,38–40,42]. These levels of force were comparable to EHTs that underwent physical conditioning of increasing intensity[41] and compared favorably to EHTs that included dermal fibroblasts[45]. To further democratize suspension culture cardiomyocyte differentiation, we extended the protocol to stirred suspension flasks and also characterized the reproducibility and

functional properties of the resulting sCMs. Moreover, stirred flask culture volumes can be readily scaled so that over a billion sCMs can be easily produced in a single run. Our hiPSC-CM bioreactor differentiation and cryopreservation pipeline and the subsequent deep morphological and functional characterization of bCMs and sCMs provides a scalable and reproducible hiPSC-CM platform to enable subsequent disease modeling and therapeutic cardiomyocyte replacement.

**Fig. 5 | Optimized cardiac differentiation protocol in spinner flasks. a** Schematic of the optimized protocol applied to magnetically stirred spinner flasks. Abbreviations as in Fig. 1. **b, c** hiPSC-CMs with low frequency of pluripotency marker SSEA4 by flow cytometry (**b**) or out of range mean EB diameter (**c**) had higher likelihood of failure, defined as <90% TNNT2+ cells (n = 8 differentiations). **d, e** hiPSC-CM yield (**d**) and purity (**e**) at bCM or sCM dd15 (bioreactor, n = 25 differentiations; sCMs, n = 6 differentiations). 2.5× and 3.8× scaling indicate 250 or 380 ml cultures, respectively. TNNT2+ cell percentage was measured by flow cytometry. **f** Spontaneous beating frequency of bCMs and sCMs at day 15 (n = number of differentiations; number of EBs: bCMs (n = 3/71); sCMs (n = 3/57). **g–n** RT-qPCR analysis of marker gene expression during bioreactor and spinner flask cardiac differentiation. Values are expressed as fold-change compared to bCMs day 5 or 15. n=number of differentiations: sCMs (n = 3). Points represent biological replicates for each independent differentiation, except for **h′** in which points represent technical replicates. **o–q** sCM Ca$^{2+}$ transients properties under 1 Hz electrical pacing 7 days after cryo-recovery. **o** Average, normalized Ca$^{2+}$ transients. **p** Maximum upstroke velocity. **q** Ca$^{2+}$ transient amplitude. Kruskal–Wallis with Dunn's multiple comparison test. n=number of differentiations/ number of wells for sCMs: cryo: n = 2/58; fresh: n = 3/40). **r–t** sCM action potential (AP) properties, optically recorded under 1 Hz electrical pacing 7 days after cryo-recovery. **r** Average, normalized APs. **s** Maximum upstroke velocity. **t** AP duration at 90% recovery (APD90). Kruskal–Wallis with Dunn's multiple comparison test. **u** Representative image of an sCM EHT. Bar, 1 mm. Spontaneously beating EHTs were serially analyzed in culture medium. **v** Spontaneous beating frequency. **w** Force generated by EHTs. **x** Time for EHT 50% contraction (C50). **y** Time for EHT 90% relaxation (R90). Data are expressed as mean ± SEM. **d–f** two-tailed Welch's unpaired *t* test. **g–n** one-way ANOVA with Dunnett's post-test. bCM and mCM data from **b–y** were replotted from Figs. 1b–f, h, i, k–p, 3l–q and 4b–e to facilitate comparisons. Source data are provided as a Source Data file.

We adapted the use of MCBs, small molecules, and optimized time points for Wnt activation and inhibition to increase efficiency in cardiac differentiation and reduced inter-batch variability. The success of these adaptations was reflected by first contractions observed as early as day 5 using our protocol (Supplementary Movie 5), 2–3 days earlier than other protocols(Supplementary Table 1)[1–5]. The bioreactor protocol yielded 1.2 million bCMs/ml, and the bCMs had functional properties that are unparalleled by other stem cell-derived cardiomyocytes. Previously reported spinner-[3] and shaker-based differentiation[7] protocols achieved 1.5–2 million hiPSC-CMs per mL, but these approaches lack reproducibility in differentiation outcomes and the functional properties of the resulting cells was not extensively evaluated. When upscaled by 3.8 fold to 380 ml, our spinner flask protocol yielded 3.4 million/ml hiPSC-CMs, at present the highest concentration reported (Supplementary Table 1)[1–7]. Whereas a prior study of cardiomyocyte differentiation scaling found less than linear increases in yield[3], we found greater than linear increases in yield at the volumes studied. Further scaling and optimization of this protocol might enable the production and cryostorage of 100 billion hiPSC-CMs, the number needed to seed artificial heart models for regenerative medicine[46]. Directions for further optimization include use of a perfusion bioreactor for hiPSC expansion prior to differentiation[26] and use of "maturation media"[40] or lactate treatment[37] to further promote cell maturity. Our studies also point out that even very similar differentiation protocols can strongly affect the functional properties of hiPSC-CMs, such as force generation in EHTs. These observations emphasize the importance of careful characterization of the functional properties of hiPSC-CMs when developing or modifying cardiomyocyte differentiation protocols.

Efficient workflows and optimal use of the large number of cardiomyocytes produced by suspension culture requires cryopreservation and cryo-recovery. A previous study pointed to line-dependent impact of cryo-recovery on the properties of hiPSC-CMs equivalent to mCMs+L[9]. Importantly, overall hiPSC-CM viability after cryo-recovery in that study was ~60%, suggesting that the cryopreservation method used was challenging to the cells. We optimized freeze/thaw procedures, including use of a controlled rate freezer and serum-free freezing reagents[5,9]. Together, these optimizations achieved 94% cell viability on cryo-recovery. Additionally, plating efficiencies for bCMs ranged from 45 to 56%, which is comparable to commercially available hiPSC-CMs. Our functional measurements indicated robust and reproducible function of cryo-recovered bCMs and sCMs, and indeed these cryo-recovered bCMs displayed features of more mature Ca$^{2+}$ transients and action potentials in line with a report by van den Brink et al., showing more ventricular subtypes in cryopreserved vs. freshly plated hiPSC-CMs[47].

Our studies clearly demonstrate the link between differentiation protocol and cardiomyocyte function. The same cell line differentiated with similar chemical cues but in different formats yielded iPSC-CMs with different functional properties. This link between differentiation protocol and cardiomyocyte function points out the need for studies introducing enhanced differentiation protocols to carefully document cardiomyocyte functional properties. Providing benchmark function parameters on calcium transients, action potentials, and contractile function will allow protocols to be compared in terms of functional properties as well as yield.

Finally, we adapted the suspension culture protocol to produce bioreactor-derived cardiac organoids (bCOs), the first cardiac organoids completely formed in a bioreactor. One bioreactor run can produce hundreds of bCOs and is not restricted by the limited throughput of multiwell dishes as reported previously. The structure of bCOs, with muscular walls surrounding a chamber and often with a muscular septum, is reminiscent of the overall organization of cardiac chambers. Our scRNAseq analysis showed that bCOs primarily contain ventricular-like CMs, as well as non-CMs such as fibroblasts and endothelial cells. However, no epicardial marker genes such as *WT1* were detected in these bCOs, consistent with distinct conditions required for their directed differentiation[48,49]. Previously published studies either co-cultured differentiated epicardial cells with COs[21] or adjusted their differentiation protocol to specifically enhance epicardial differentiation on the surface of COs by CHIR supplementation 7 days after mesoderm induction[20] or continuous retinoic acid supplementation[24]. Application of these methods and further optimization in the future will enable the formation of bCOs with greater cellular diversity. We project uses for bCOs as microphysiological systems to characterize tissue level cardiac responses to disease-causing mutations or therapeutic manipulations, and to assess the impact of congenital heart disease associated variants on cardiac development.

# Methods

### Ethics declaration

Studies with patient-derived materials were conducted under protocols approved by the Boston Children's Hospital Institutional Review Board. Patients' informed consent was obtained to participate in this study.

### Human pluripotent stem cell culture

WTC-11 is a wild-type human male iPSC line (Coriell Institute: # GM25256). WTC-Cas9 (Control) was derived from WTC-11 by inserting CAG-rtTA::TetO-Cas9 (Addgene #73500)[50] into the AAVS1 locus to yield a dox-inducible Cas9 hiPSC line. This control hiPSC line underwent cardiac differentiation in parallel in the bioreactor and in monolayer format followed by functional experiments and comparison to bioreactor- and spinner-differentiated hiPSC-CMs (bCMs, sCMs) was done using only the Ctrl line. Additionally, control hiPSC were used for the generation of bioreactor-derived cardiac organoids (bCOs). Genome integrity was tested by digital karyotyping on the

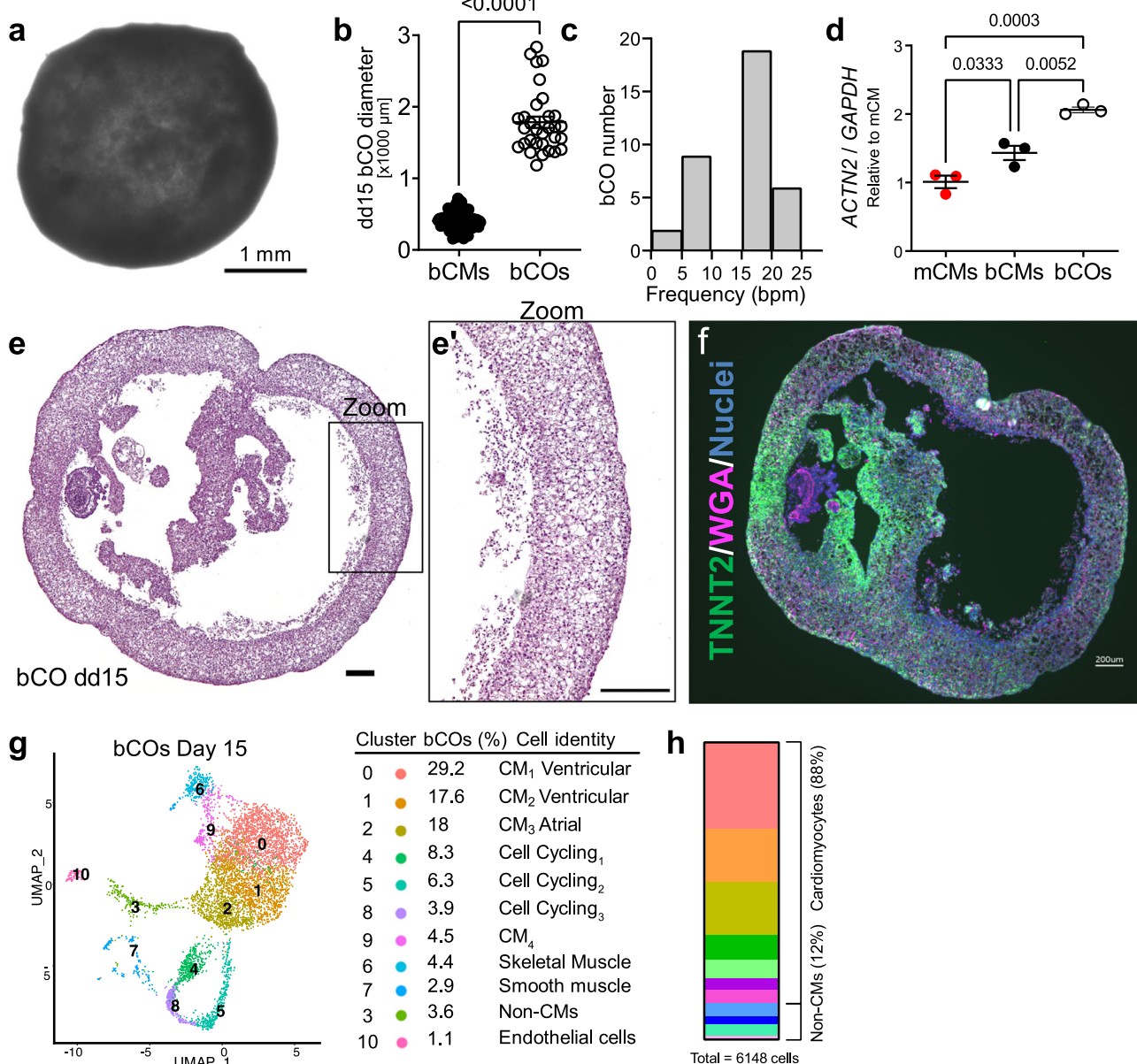

**Fig. 6 | Generation of bioreactor-derived cardiac organoids (bCOs).**
**a** Representative image of a bCO at dd15. **b** EB diameter of dd15 bCMs (40 EBs from three differentiations) and bCOs (31 bCOs from two differentiations). Two-tailed Welch's unpaired *t* test. **c** Spontaneous beating frequency of dd15 bCOs (36 bCOs from three differentiations). **d** Expression of cardiac marker gene *ACTN2* in dd15 mCMs, bCMs, and bCOs. *ACTN2* transcript level was measured by RT-qPCR. One-way ANOVA with Sidak's post-test. Points represent biological replicates for each independent differentiation. **e** Hematoxylin and eosin staining of a bCO section

dd15. Boxed area is enlarged in **e´**. Scale bar in **e** and **e´** = 200 μm. **f** bCO section stained with TNNT2 antibody, wheat germ agglutinin (WGA), and nuclei (Hoechst). Scale bar = 200 μm. **g**, **h** Cellular composition of dd15 bCOs determined by scRNAseq followed by UMAP clustering (**g**). Stacked bar graph (**h**) of the percentage of each cell state in bCOs. Cell states were grouped into cardiomyocytes (top) and non-CMs (bottom). Data are shown as mean ± SEM. Non-CMs, non-cardiomyocytes. Source data are provided as a Source Data file.

nanostring platform (Nanostring Technologies; Supplementary Fig. 1b) and previously published ichorCNA method[51]. Additionally, cell culture supernatants were routinely screened for mycoplasma contamination using the LookOut® Mycoplasma PCR Detection Kit (Sigma, # MP0035-1KT). Furthermore, genetic mutations modeling cardiomyopathies were introduced into the WTC-Cas9 (WTC-Cas9-edited #1-8) and the ctrl line PGP1 (PGP1-edited #1) using Cas9 genome editing, as described previously[52]. Patient-derived lines (1, 2, 3 and 4) were reprogrammed from blood using the CytoTune™-iPS 2.0 Sendai Reprogramming Kit (Thermo Fisher Scientific, # A16517). These cells were used for bioreactor and/or spinner flask-based differentiations.

**Master cell banks**

Master cell banks (MCBs) were established as described previously[25]. In short, hiPSCs were grown and massively expanded in T80 cell culture flasks (Life Technologies, #178905; see details below). Expanded hiPSCs were dissociated and all resulting cells were frozen in mFreSR™ (STEMCELL Technologies, #05854) at 3 million hiPSC per cryovial, whereby 4x T80 flasks yield-60 million hiPSCs. These cells were tested for karyotype abnormalities, genotyped and screened for mycoplasma contamination. Massive expansion of hiPSCs derived from the MCB was repeated to generate a working cell bank. For subsequent cardiac differentiation and functional analysis, solely cells derived from MCBs and working cell banks were used to

increase reproducibility between independent differentiation batches.

## Generation of hiPSC-CM in 2D monolayer culture

Monolayer cardiomyocyte differentiation was induced as previously described[2] with some minor modifications (Supplementary Fig. 1a'). Briefly, Ctrl cells were washed once with PBS, detached by incubation with Versene (Invitrogen) for 5–10 min and seeded onto 12-well plates (Fisher Scientific, #0877229) pretreated with 1:100 (v/v) diluted Geltrex (Life Technologies, #A1413302) at a density of ~45,000 cells/cm2 in E8 medium supplemented with 10 µM of ROCK inhibitor Y-27632 (R&D, #1254). When cell confluency reached about 50–60%, cells were treated with basic medium (RPMI-1640, #61870127) plus 1x B27 minus insulin (#A1895601, all Life Technologies) containing 7 µM CHIR99021 (STEMCELL Technologies, #72054) for 48 h, followed by basic medium containing 5 µM IWR-1-endo (STEMCELL Technologies, #72564) for another 48 h. On day 7 of differentiation cells were cultured in basic medium supplemented with 1:1000 (v/v) insulin (Sigma-Aldrich, #I9278) until day 15. For lactate treatment[37], cells were fed with 5 mM lactate (Sigma-Aldrich, #L7900) in basic medium without B27 for 4 days starting from day 10 after differentiation and then switched back to basic medium plus B27 and Insulin at day 14 of differentiation. Finally, cells were dissociated for 1–2 h depending on cell density using Collagenase II (Worthington, #LS004176) as described previously[5] and below. Single cell suspensions were frozen in STEMdiff™ Cardiomyocyte Freezing Medium (STEMCELL Technologies, # 05030) using a controlled rate freezer (Grant, CRF-1) and stored in liquid nitrogen as described below.

## Generation of hiPSC-CMs in suspension culture systems

All bioreactor preparation steps were performed as described previously for the DASGIP® Parallel Bioreactor Systems (Eppendorf) with some minor modifications[4]. Inner vessel walls of the bioreactor (250 mL total volume) were coated with 1% (wt/vol) Pluronic F-127 (Sigma-Aldrich, # P2443) in PBS (Life Technologies, # 10010049) using a total volume of 100 mL for 1 h at 37 °C. Coated vessels were washed three times with distilled water. In parallel, calibration of the pH sensor was performed at pH 4 (ACROS, #61104-5000) and 7 (Fisher, # SB107-500) using commercially available buffer solutions. Upon completion all parts of the bioreactor were assembled (glass vessel, dissolved oxygen (DO) sensor, pH sensor, and head caps) and autoclaved at 121 °C for 15 min. The sterilized bioreactor vessel was put under a laminar flow and 100 mL of PBS were added to each of the 250 mL volume capacity glass vessel (DASbox® Vessel, SR0250ODLS, 60–250 mL, overhead drive, 8-blade impeller (60° pitch), #76SR0250ODLS). The vessel was placed onto the temperature controller and calibration of DO sensor was performed under agitation (60 rounds per minute; rpm) with 21% $O_2$, 5% $CO_2$ at 37 °C and 10 standard liters per hour (sL/h) overlaying flow gas. After 15 days of cardiac differentiation (described below) the bioreactor set-up was disassembled and cleaned as described previously[4].

Bioreactor cardiomyocyte differentiation was induced as previously described[2] with modifications (Fig. 1a; Supplementary Fig. 1a). At the start, hiPSCs were cultured in T80 cell culture flasks pretreated with 1:100 (v/v) diluted Geltrex (Life Technologies, #A1413302) incubated in a humidified incubator at 37 °C, with 5% $CO_2$ at a density of 15,000 cells/cm². Cells were maintained in E8 medium (Life Technologies, #A1517001) with daily medium change until 90% cell confluency was reached. For dissociation, T80 flasks were washed once with PBS, incubated with 5 mL Versene (Life Technologies, #1540066) for 15 to 20 min at 37 °C and dissociation was stopped by adding 5 mL E8 medium supplemented with 10 µM of ROCK inhibitor Y-27632. On average one flask resulted in 10 to 20 million hiPSCs depending on cell line and confluency upon dissociation. Cells were collected in 50 mL Falcon tubes and counted manually using a Neubauer Hemacytometer

(Weber Scientific, #3048-12). After centrifugation (200 × $g$, 5 min), 50 million single cells were resuspended in 50 mL E8 medium supplemented with 10 µM of ROCK inhibitor. The bioreactor vessel was taken from the bioreactor system and placed under the laminar flow. The PBS solution in the vessel was aspirated and replaced with 50 mL E8 (+10 µM Y-27632) and 50 mL of the hiPSC solution resulting in a final volume of 100 mL per vessel. Cells were agitated at a speed of 60 rpm, gassed with 21% $O_2$ and 5% $CO_2$ by 10 standard liters per hour (sL/h) overlay gassing and maintained at 37 °C. pH was maintained at 7. The next day diameter of spontaneously formed embryoid bodies (EBs) was measured to estimate time of differentiation start. EB size was assessed by averaging at diameter measurements of at least 50 independent EBs. If critical diameter (100–300 µm) was reached, cardiac differentiation was induced by a complete change of the medium to RPMI 1640 with B27 (w/o Insulin; basic medium) supplemented with 7 µM CHIR99021 (day 0). After 24 h (Day 1), the complete medium was changed to basic medium and cells were incubated for an additional 24 h. On day 2, the complete medium was changed again to basic medium containing 5 µM IWR-1-endo for 48 h. If oxygen consumption was high (>30% XO2.PV) a 50% medium change using basic medium containing 5 µM IWR-1-endo was done. On day 4, the complete medium was changed to basic medium and cells were incubated for an additional 72 h with an optional 50% medium refreshment either on day 5 or 6. From day 7 on cells were cultured in basic medium supplemented with 1:1000 (v/v) insulin and 50% medium was refreshed on day 9, 11, 14. If oxygen consumption was high (>30% XO2.PV) a 50% medium change was performed every day until dissociation. Finally, cells were enzymatically dissociated on day 15 for 3–4 h depending on EB size and density, and frozen using a controlled rate freezer as described below.

Besides the bioreactor system we used a spinner flask-based suspension system (Pfeiffer CELLspin Spinner System, Cole Palmer®; CELLSPIN CONTROL UNIT #501960613; CELLSPIN STIR PLATFORM 4 POS #501960614). This system is placed into the incubator without real time monitoring or adjustment of pH, oxygen, temperature, and $CO_2$ For our standard differentiation and our 2.5x scaling experiment we used 250 mL spinner flasks for cardiac differentiation (FLASK SPIN 250 ML W/2 STIRRERS #501960611). For the 3.8x scaling experiment we used a 1 L spinner flask (FLASK SPIN 1 L W/2 STIRRERS #501960612; All spinner flask-based components were ordered through Fisher Scientific). There is no need for calibration of this system. Autoclaved spinner flasks were pretreated with 1% Pluronic as described above and prepared for cell inoculation. The same cardiac differentiation protocol as described above was applied to the spinner flasks (Fig. 5a, Fig. S1a). The only differences to the bioreactor system were seen for EB sizes, which reached an optimal diameter for CHIR incubation ranging from 80–100 µm and low buffering capacity due to missing pH monitoring and adjustment. Therefore, medium in spinner flask cardiac differentiations had to be replaced at least once a day until day 15. Dissociation and freeze/thaw procedures were applied to spinner flask-differentiated hiPSC-CMs as described below for bioreactor-differentiated hiPSC-CMs.

## Dissociation of hiPSC-CMs using Collagenase II

On day 15 monolayer and suspension differentiated hiPSC-CMs were dissociated using Collagenase II solution containing HBSS (Gibco, # 14175-095), Collagenase II (200 units per ml, Worthington, #LS004176), HEPES (10 mM, Sigma, # H3375), ROCK inhibitor Y-27632 (10 µM), and n-benzyl-p-toluenesulfonamide (BTS, 30 µM, VWR, #TCB3082)[5]. Monolayer differentiated hiPSC-CMs were washed twice with pre-warmed HBSS and incubated in Collagenase II solution (0.5 mL per 12-well) at 37 °C in 5% $CO_2$ for 1–2 h until single cells were apparent. For dissociation of suspension-derived hiPSC-CMs the bioreactor was taken from the temperature controller and opened under the laminar flow. The vessel was mixed manually to distribute all EBs equally in the vessel and 10 mL were taken out of the bioreactor

and transferred into a 15 mL Falcon tube to estimate EB volume as described previously[5]. Approximately, 12 mL of Collagenase II solution were used for 200 µL EB volume. Once EB volume was established, all EBs were pooled in a 50 mL Falcon tube and washed twice with HBSS, resuspended in Collagenase II solution, transferred to T175 suspension flasks (Sarstedt, #83.3912.502) pretreated with 1% (wt/vol) Pluronic F-127, and incubated at 37 °C in 5% $CO_2$ for 3–4 h. Dissociation times vary, therefore, it is recommended to check after 1 h regularly on the progression by carefully tapping culture plates or flasks on a hard surface. If EBs were dispersing into single cells or very small clusters, they were gently triturated 5–10 times in their flask and transferred to a 50 mL Falcon tube. Dissociation reaction was stopped by adding the same volume of blocking buffer to the single cell suspension, which contained RPMI-1640 w/o B27 (Sigma, #D8764), with 6 µL DNase II (Sigma, #D8764) per mL RPMI-1640 and 10 µM of ROCK inhibitor Y-27632 (R&D, #1254). Monolayer and suspension-differentiated hiPSC-CMs were counted, centrifuged ($200 \times g$, 5 min) and prepared for downstream application or freezing (see below).

## Freezing and thawing of hiPSC-CMs
For freezing, pelleted hiPSC-CMs were resuspended in cold (4 °C) STEMdiff™ Cardiomyocyte Freezing Medium (STEMCELL Technologies, # 05030) and 1 mL was aliquoted into cryotubes (Fisher Scientific, #12565163 N) with cell numbers ranging from 1 to 80 million cells per cryotube. Cryotubes were frozen to −80 °C in 1 h using a controlled rate freezer (Grant, CRF-1; Supplementary Table 5) followed by transfer to liquid nitrogen tanks (−150 °C) for long-term storage.

For thawing of hiPSC-CMs a cryotube was removed from −150 °C storage and immediately placed in a water bath set at 37 °C. Constant moving of the cryotube in the water assured optimal distribution of heat and therefore uniform thawing of frozen cells. As soon as no ice crystal was visible cryotubes were placed under laminar flow and cell suspension was gently transferred to a 50 mL Falcon tube using a 1-ml pipette. The empty cryotube was rinsed with 1 mL of RPMI-1640 (room temperature) w/o B27 supplemented with 10 µM of ROCK inhibitor Y-27632 (R&D, #1254) to recover residual cells, followed by dropwise addition to the 50-ml Falcon tube containing the cell suspension over 90 sec under gentle swirling. Next, 8 mL of RPMI-1640 (room temperature) w/o B27 were added to the cell suspension, whereby the first mL was added dropwise over 1 min and remaining 7 mL were added over 30 sec under gentle swirling. Finally, cell suspension was inverted three times in the 50 mL Falcon tube, viability and cell count was determined using Trypan Blue (Sigma-Aldrich, #T10282) and a hemocytometer (Sigma-Aldrich, #Z359629), and centrifuged at $200 \times g$ for 5 min at room temperature. Pelleted hiPSC-CMs were resuspended and prepared for downstream applications as described below. Plating efficiency was determined by seeding bCMs and mCMs in 12-well plates at different densities (Supplementary Fig. 1g). 24 h later cells were dissociated with Accutase (STEMCELL Technologies, #07922) for 5–10 min, hiPSC-CMs were pooled, and cell count was determined using Trypan Blue and a hemocytometer.

## Flow cytometry
$1 \times 10^6$ single cells were added into a 15 mL Falcon tube, washed with 5 mL PBS, and centrifuged at $200 \times g$ for 5 minutes. The supernatant was discarded and cells fixed for 10 min in 4% PFA, for intracellular epitopes, or ice-cold methanol (−20 °C), for extracellular epitopes, followed by centrifugation at $200 \times g$ for 5 minutes. Fixed cells were resuspended in 500 µL permeabilization buffer containing PBS, 5% (v/v) FCS/FBS (R&D systems, #S11150), 0.5% (w/v) Saponin (Sigma, #47036-50G-F) and 0.05% (w/v) Sodium azide (Sigma, #S2002) and incubated overnight or 1 h at 4 °C. Cell suspension was washed with 5 mL PBS, and centrifuged at $200 \times g$ for 5 minutes. Pelleted cells were resuspended in permeabilization buffer containing directly labeled antibodies as shown in Supplementary Table 3 and incubated

for at least 45 min at 4 °C in the dark. Cells were washed two times with 5 mL of PBS, the final pellet was resuspended in 200 µL PBS and cells analyzed by flow cytometry (LSR Fortessa Analyzer, BD Biosciences).

## Gene expression
Total RNA from Cells, previously stored at −80 °C, was extracted using an RNeasy Plus Universal Mini Kit (Qiagen, Valencia, CA, USA). The RNA integrity was assessed by automated electrophoresis using the RNA ScreenTape Analysis and 4200 TapeStation System (Agilent), concentrations were measured using a Nanodrop 8000 spectrophotometer (Thermo Scientific, Wilmington, DE, USA), and RNA was stored at −80 °C. Synthesis of cDNA was performed with 0.5 µg total RNA using a High Capacity cDNA Reverse Transcription Kit (Applied Biosystems, Foster City, CA, USA), according to the manufacturer's protocol in a MyCycler Thermal Cycler (Bio-Rad, Philadelphia, PA, USA). The cDNA was obtained in a final volume of 20 µL and stored at −20 °C until it was used for the RT-qPCR expression assays. RT-qPCR was performed on the following genes using the SYBR green assay (Life Technologies #4368708). PCR assays were carried out in 384-well plates using a CFX384 Touch Real-Time PCR Detection System (Bio-Rad). Relative expression was calculated using the 2-ΔΔCT method[53], and results are presented as fold-change versus the control group mean values, normalized to GAPDH. Supplementary Table 4 shows all primers used in this study.

## Protein quantification
In all, 30 µg cell lysates[54] were separated by 4–12% SDS-PAGE (Invitrogen, #NW04120BOX) and transferred to 0.45 µm nitrocellulose membranes (Bio-Rad Laboratories, #1620115). The membranes were blocked with 5% BSA and incubated with primary antibody overnight at 4 °C, followed by three consecutive TBS-T washes and incubation with secondary antibody for 1 h at room temperature (Supplementary Table 3). Western blot signals were captured using a Fujifilm LAS-3000 imager and quantified using Fiji.

## Single cell RNA sequencing
Collagenase II dissociated hiPSC-CMs on day 15 of cardiac differentiation and counted using a hemocytometer. Cells were resuspended in PBS + BSA (0.04%). Libraries were generated using the Chromium platform (10x Genomics) with the Next GEM Single Cell 3′ Reagent Kit v3.1, using an input of 1 million cells per mL. Gel-Bead in Emulsions (GEMs) were generated on the sample chip in the Chromium controller. Barcoded cDNA was extracted from the GEMs by Post-GEM RT-cleanup and amplified for 12 cycles. Amplified cDNA was then fragmented and subjected to end-repair, poly A-tailing, adapter ligation, and 10x-specific sample indexing following the manufacturer's protocol. Libraries were quantified using the TapeStation (Agilent Technologies). Libraries were sequenced using NextSeq 500 (Illumina) at Harvard Medical School. Downstream differential expression and clustering analysis was performed using the Seurat V.4.0 package, as described in the tutorials (http://satijalab.org/seruat/). CellRanger matrices were imported for each sample with default parameters, and distributions of gene number, UMI number and % mitochondrial gene expression were examined for each sample to filter out cells of low quality. Cells with greater than 20% of genes coming from mitochondrial genes were selected against, as well as those with fewer than 200 genes or 400 UMIs. The doublets were detected and removed using DoubletFinder (v2.0.3). The resulting subset Seurat objects were normalized using the scTransform workflow and further scaled and normalized the RNA assay in order to perform downstream differential expression analysis and marker visualization utilizing the FindMarkers and FeaturePlot functions on the RNA assay. Uniform manifold approximation and projection (UMAP) was performed and iteratively modified after performing

marker gene expression and examining expression of key markers. Cell scores for chamber-specific gene expression and CM maturation were determined by evaluating the average expression of a set of representative marker genes for each metric.

## Immunostaining of unpatterned and micropatterned hiPSC-CMs

iPSC-CMs were thawed and cultured for 7 days in unpatterned 96-well plates (μclear®, Greiner Bio-One, 655090) and subsequently prepared for immunofluorescence analysis, as described previously[18]. The primary antibodies ACTN2 (Miltenyi Biotec, 130-119-766) and Vimentin (R&D Systems, MAB2105) were used followed by the secondary antibody anti-mouse Alexa Fluor® 488 (Life Technologies, LT A11029) and Alexa Fluor® 555 (Life Technologies, A21434) (Supplementary Table 3). Nuclei staining was obtained with Hoechst 33342 (5 μg/ml, Thermo Fisher Scientific). Images were obtained by confocal microscopy using a Zeiss LSM 800 or Olympus FV3000R confocal microscope with a ×40 or ×60 oil immersion objective.

Micropatterns were produced on glass coverslips (12 mm, VWR CAT NO 48366-252) coated with a 1:1 ratio of Polydimethylsiloxane 184 (10:1 Elastomer Base: Curing Agent, by The Dow Chemical Company LT H047M73001) and 527 (1:1 Part A: Part B, The Dow Company LT H047L6A013). Coverslips were coated for 46 hours in a 65 °C oven. Then stamps with features of islands containing 7:1 aspect ratios[32] were coated for 1 hour with Fibronectin (50 μg/ml) diluted in Geltrex (1:200, Life Technologies, # A1413302). In the meantime, PDMS-coated coverslips are exposed to UV Ozone for 8 minutes and the patterning process is performed by placing the dried stamps onto the coverslips. 1% Pluronics F-127 (Sigma-Aldrich, #P2443) is used to wash the coverslips for not more than 10 minutes, followed by washing 3 times with room-temperature PBS. To assess hiPSC-CMs in micropatterns 50,000 cells were plated on one micropatterned glass slide with a 7:1 size ratio and cultured for 1, 3 and 7 days. For immunofluorescence analysis cells were fixed with 4% PFA for 10 min at 4 °C, followed by permeabilization in 0.1% Triton X-100 (Sigma-Aldrich, LT 069K0049) dissolved in PBS for 10 min at RT. hiPSC-CMs were incubated in blocking buffer (PBS, 3% BSA (Sigma, LT SLBW9820)) for 1 hour at room temperature and then incubated with conjugated antibodies for α-actinin/FITC (1:50, Supplementary Table 3) and Phalloidin 647 (1:400, Supplementary Table 3) in addition to Hoechst (1:500) in staining buffer (PBS, 1% BSA (Sigma, LT SLBW9820)) for 2 hours at RT. Micropatterned glass slides were then mounted using Prolong Diamond Antifade Mountant (Invitrogen, LT 2273639). Images were obtained by confocal microscopy using a Olympus FV3000R confocal microscope with a ×60 oil immersion objective.

## Morphological analysis of hiPSC-CMs

Quantification of myofibrillar disarray and cell area was evaluated with Fiji (ImageJ) as described previously[15,18]. Additionally, values for circularity were captured with Fiji (ImageJ) by assigning a cell and using the "Measure" function. To assess sarcomere alignment in micropatterned hiPSC-CMs we used previously reported methods from the Disease Biophysics Group[33] that utilizes ImageJ Plugins (OrientationJ) and a custom-made Matlab script for structural analysis of the cells Resulting parameters for sarcomere alignment referred to as "Orientational Order Parameter" (OOP) were captured, whereby the OOP2 value corresponded to the orthogonal alignment of actin bundles representing Z-disc regularity. Multinucleation was assessed by manually counting the number of nuclei per cell in unpatterned and micropatterned formats. Morphology of nuclei was assessed using a previously published Fiji plugin[34].

## Ca²⁺ and voltage imaging

For high throughput $Ca^{2+}$ and voltage imaging, we used the Vala Biosciences Kinetic Image Cytometer (KIC). Cryopreserved and/or

fresh hiPSC-CMs were seeded onto polystyrene 96 well plates (Greiner Bio-One, 655090) at 100,000 cells per well. After four or 7 days in culture, cells were stained with 0.1 μg/ml Hoechst 33342 (Life Technologies, H1399) and 2.5 μM Fluo-4 calcium indicator dye (Invitrogen F14201) or 1:1000 FluoVolt membrane potential kit (Invitrogen F10488) in Tyrode's solution for 30 min at 37 °C. Cells were then washed with Tyrode's solution and imaged by KIC with a ×20 objective at 67 frames per second. The imaging protocol consisted of 1 Hz electrical stimulation with 10 seconds of pre-pacing followed by 10 seconds of video acquisition. Raw data was analyzed with CyteSeer software (Vala Biosciences) using custom scripts for analysis of calcium transient kinetics and action potential duration.

## Seahorse measurements

For assessment of mitochondrial function, we used the Seahorse XF Cell Mito Stress Test Kit (Agilent #103015-100). Cryopreserved or freshly plated hiPSC-CMs were seeded onto Seahorse XF96 cell culture microplates (Agilent #101085-004) at 50,000 cells per well. After 7 days in culture, materials, media and cells were prepared according to the manufacturer's protocol and the Seahorse XF cell mito stress test default protocol was run in Seahorse XF Pro Analyzer (Agilent). Data was analyzed using the Seahorse Wave Controller software (Agilent) and normalized to cell density using the CyQUANT Direct Cell Proliferation Assay (Life Technologies # C35011).

## Generation of engineered heart tissues

Engineered heart tissues (EHTs) were generated as described previously[5,55] with some minor modifications. Briefly, $0.8 \times 10^6$ hiPSC-CMs were used to generate each EHT. Cells were transduced with adenovirus ChR2-YFP on the day of casting or after 7 days in vitro. We modified the standard EHT culture medium (EHT-medium in the referenced literature[5]) by replacing DMEM with RPMI 1640 plus B27 minus insulin, removing 10% heat-inactivated horse serum, and reducing aprotinin concentration to 5 μg/ml. This resulted in EHTs initiating contraction as early as day 1–3 after EHT assembly, versus day 7–10 reported in the literature[5]. EHT contraction was recorded as described below from day 7 on and functional analysis was performed from day 27 to day 33. For immunohistochemical analysis EHTs were cryosectioned and stained with primary antibodies shown in Supplementary Table 3. Sections were imaged using an Olympus FV-3000 confocal microscope or Zeiss LSM 880 and analyzed with Fiji (ImageJ). Similarly, bCOs were cryosectioned for immunohistochemical analysis (Supplementary Table 3) and paraffin sections were prepared and stained with hematoxylin and eosin (Sigma), and masson's trichrome (Sigma) according to the manufacturer's protocol.

## Functional assessment of EHTs

EHTs in a 24-well plate were placed in a stage-top incubator and maintained at 37 °C, 5% $CO_2$. EHTs were optically paced at different frequencies using blue LEDs positioned above the plate, and recorded from below at 30 frames per second through a 561 nm long-pass filter (Semrock BLP02-561R-32) using an 8 mm f/1.4 lens (ThorLabs MVL8M1) mounted on a Basler acA1920 camera. EHT post movement was tracked post-hoc using the multi-template matching FIJI plugin[56]. Twitch force measurements were subsequently measured by applying post deflection to the beam bending theory for a known Young's modulus of the posts, as described in detail elsewhere[57].

## Statistics

Group data are presented as mean ± SEM. GraphPad Prism 10.0.2 (GraphPad Software, San Diego, CA, USA) was used for data analysis. Data was tested for normal distribution, if criteria were not met Kruskal–Wallis with Dunn's multiple comparison test was used. If data was normally distributed, we used unpaired Student's $t$ test with Welch's correction, one-way ANOVA, or two-way ANOVA followed by

Sidak's or Tukey's post-test, depending on sample size and comparisons and as specified in each figure legend. Chi-square analysis was performed to compare the distribution of categorical data. A $p$ value $< 0.05$ was considered statistically significant.

**Reporting summary**

Further information on research design is available in the Nature Portfolio Reporting Summary linked to this article.

## Data availability

Datasets, analysis, and study materials will be made available on request to other researchers for the purpose of reproducing the results or replicating the procedures. The scRNAseq data generated in this study have been deposited in the Gene Expression Omnibus database under accession code GSE263372). Source data are provided in this paper.

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

## Acknowledgements

The authors would like to thank Feng Xiao for providing hiPSCs for cardiac differentiation and Suellen Lopes Oliveira for assistance in graphic design. WTP and KKP were supported by the NCATS Tissue Chips Consortium (UH3 HL141798 and UH3 TR003279), R01HL163937, and charitable support from the Boston Children's Heart Foundation. W.T.P. and M.P. were supported by funding from Additional Ventures.

## Author contributions

M.P. and W.T.P. conceived of the project and designed the experiments. M.P., P.B., M.A.T., Y.T., K.S., R.H.B., M.E.S., J.M., D.Y., A.M.C., B.G., C.H., N.J.A., F.N., J.C., J.Mi., D.W., Y.Z., F.L., and X.L. conducted the experiments and analyzed the data. K.K.P. and V.J.B. contributed reagents and resources. M.P. and W.T.P. interpreted the data and wrote the manuscript.

## Competing interests

The authors declare no competing interests.
