## [Peer Review File · Nature Communications]

REVIEWER COMMENTS

Reviewer #1 (Remarks to the Author):

The paper by Prondzynski et al described a method which used a stirred suspension bioreactor to produce on average 124 million hiPSC-CMs with >90% purity using a variety of hiPSC lines (19 differentiations; 10 iPSC lines).

The novelty of the study is weak. The methods using bioreactors to differentiate human pluripotent stem cells not new, as several other groups have reported already (doi:10.1038/s41596-019-0189-8; doi.org/10.1172/jci.insight.99941; doi: 10.3389/fbioe.2021.674260). The only difference may be the CMs generated each batch is larger in the current paper as compared with others.

3D bioreactor promotes maturation of CMs is already known. Authors shall provide detailed analysis of structure, like EM, ion channel, electrophysiology, and metabolism to show that their method is better than others.

One issue is the 10 hiPSC lines: as shown in Fig 1B, did the first 5 hiPSC lines originate from the same one: WTC-Cas9? If so, probably they share the similar genetic and epigenetic background even after CRISPR edited

Reviewer #2 (Remarks to the Author):

The paper describes an optimized method for producing high numbers of hiPSC-cardiomyocytes at high purity. Whilst multiple methods have been described for this previously, many of these are used by companies and are not openly accessible to the research community. This method appears robust and is claimed to be reproducible across multiple (10) hiPSC lines and differentiations although not all claims are supported or documented by the data presented, particularly reproducibility between lines. However, the functional properties of the derivative hiPSC-CMs are at least equal to or better than those described for standard monolayer differentiation which is encouraging. There are a number of questions that need to be addressed if the protocol is to be of real value to the field.

Major points:

1. To what extent is the protocols dependent on this particular brand of bioreactor? It is widely known that bioreactors differ in their properties and in general each type needs optimising. The

timings in the protocol, cell seeding densities etc may be dependent on the bioreactor properties. If it is not possible to test the protocol using another brand, then the authors could discuss how long it took to optimise this protocol, which parameters they changed etc. This would be of value to users starting with different bioreactor platforms.

2. It would be helpful to show in each of the legends which hiPSC line is exactly being used. Most seem to have been done with the WCT-Cas9 line with or without the TNNI1-GFP reporter. For example, in the first section of the results “Optimized bioreactor differentiation protocol” there seems to be no reference to the other 10 lines tested. The same is true for the others sections: all seem to have been carried out using the WCT line.

3. Could the authors explain the higher degree of maturity in the bCMs versus the mCMs. Was this related to the other (non-cardiomyocyte) cell types present? Eg endothelial cells versus fibroblasts?

4. Regarding the inability of mCMs to follow pacing at 1Hz: this seems somewhat strange as many reports show that pacing is possible. Do the authors have an explanation?

5. Regarding sarcomere length in mCMs versus bCMs, the difference between 1.6um and 1.7um (fig4j) seems very small to claim they were different: was it statistically significant over many EHTs measured at multiple positions in the EHTs? Eg were the measurements made in the middle or near the poles of the EHTs? Examining 3 and 2 EHTs respectively seems on the low side.

6. Loss of functional properties after cryopreservation (ref #9): is this the case here? Not all studies have reported cryopreservation is detrimental to hiPSC-CMs (e.g., DOI: 10.1016/j.scr.2019.101698). This claim needs to be supported by head-to-head comparisons of single parameters on the same CM batch before and after cryopreservation. Which functional parameters changed and which did not?

Minor points:

1. Abstract: hiPSC-CMs have not actually yet proved to be valuable for cardiac regeneration. This is not then limiting clinical translation in regenerative medicine at least. Please re-word.

2. Please be consistent in use of hiPSC-CM versus iPSC-CM

Reviewer #3 (Remarks to the Author):

The manuscript by Prondzynski and colleagues describes a bioreactor protocol for the generation of cardiomyocytes from human iPSCs. They compare the bioreactor CMs (bCMs) to CMs they generate by monolayer differentiation using a protocol based on the bioreactor conditions. They claim that it is more efficient to generate bCMs and that these bCMs are more mature than monolayer CMs.

As the authors acknowledge there have been a number of publications over the last few years describing the generation of CMs using bioreactors, which in the authors' opinion these have been plagued by issues in terms of inconsistent differentiations. Although they claim to have addressed this reproducibility issue by generating bCMs from 10 different iPSC lines, from the data provided it is difficult to be convinced of this.

How many differentiations were performed in total per line? For several of the lines it appears that only 1 differentiation was performed, making it difficult to evaluate the reproducibility. Indeed for patient#4 it does not appear that they managed to successfully differentiate it. Additionally 50% of the lines are sub clones from the same parental line (WTC) - these cannot be considered completely independent lines. It is also not clear how the MCBs used by the authors differ compared to how researchers usually stockpile and characterise their undifferentiated cells. More details regarding this need to be provided.

The authors also link the SSEA4% and size of the EBs to whether a bioreactor run is likely to be successful, but it is not obvious how predictive this is as one-third of the failed runs had an SSEA4 % above the 70% cutoff. Likewise, while they claim that EB diameter should be between 100-300um, more than half of the failed runs had EBs within that range.

Likewise for the comparison to the monolayer differentiation, it is not clear how frequently this was performed and appears to have only been done on the one cell line that failed the most in the bioreactor differentiations. Therefore when comparing yield for example (fig 1e) it should not be compared to other independent cell lines which clearly were the better performing ones in the bioreactor. It is also not obvious what steps the authors undertook to optimise the monolayer differentiation conditions. Seeding density is known to be a key factor in influencing the differentiation efficiency. As well it cannot be assumed that the same concentration of the small molecules used for the bioreactor differentiations will be optimal for the monolayer differentiations.

Similarly the conclusions made from the qPCR and scRNAseq are questionable. The larger inter-batch variation in ACTN2 levels seen in the mCMs is due to this gene not detected at all in one batch of mCMs - indicating that either the authors are including failed monolayer differentiations which produced no CMs in their samples (which should then also apply to the bCMs), or there was a technical issue with those measurements. The authors also highlight the higher expression of COL1A1 and COL3A1 in the mCMs at d5-7, but do not comment that by d8 expression levels of these genes are either the same or lower than what is observed in the bCMs, questioning whether this does reflect a greater proportion of non-CMs. The authors also note that the differentiation appears to be delayed in the monolayer vs bioreactor differentiations. It is also surprising that from

the scRNAseq only 50% of the monolayer-differentiated cells are classified as CMs, despite cTnT FACS indicating the purity to be ~75-80% based on fig 1f. How do the authors explain this especially when the correlation between cTnT and scRNAseq-predicted CM content for the bCMs is much similar.

The authors also state that they have managed to solve various issues surrounding the cryopreservation of hPSC-CMs, in particular those surrounding the percentage of CMs that are recovered and functional impact that freezing might have on the CMs. Whether cryopreservation does in fact have a negative functional impact remains debatable. The authors have referenced an article that did observe this, however other publications (e.g. Miller et al Curr Protoc Stem Cell Biol; van den Brink et al Stem Cell Res) do not observe such issues. Likewise many companies are providing cryopreserved hPSC-CMs, which can be paced. The percentage of CMs recovered that the authors report (>90%) is much higher than what others have observed, but how this was determined and at which point in the thawing procedure is not described in the paper. Likewise details on the optimisations they performed related to cell dissociate and cryo-protectant media are not described. The freezing density also varies greatly (between 1×10^6 - 20×10^6 /ml). Did this have any influence on recovery or functionality?

Lastly, they also observe a higher percentage of multi-nucleated mCMs compared to bCMs, and comment that multinucleation tends to indicate cardiac disease. This % of multi-nucleated CMs, even for the bCMs is much higher than what is observed in other publications. Indeed in the Mosquera paper that they cite, for the disease CMs these values were less than 2%. Similarly the authors believe the mCMs show a higher degree of DNA damage following thawing. What do they think the reason for this is, and have they ruled out that this is not due to the cell line used for the differentiations modified to be able to express Cas9. Again use of the original WTC1 cell line here would have been better.

Minor comments:

The authors should provide more details regarding the bioreactor setup they established. What was the rationale of using Rushton-type impellers (typical use for fermentations)? How was pH controlled and adjusted? What were the target, upper and lower limits? They also need to provide more details regarding the media change at day 3. They mention high O₂ consumption, but how was this determined by medium discolouration. It is not clear what the parameters for performing medium exchanges were.

Table S1 - data regarding the ventricular CMs appears to be missing.

Fig 1a - maintenance is spelt incorrectly

Reviewer #4 (Remarks to the Author):

Prondzynski and colleagues present a study on stirred bioreactor cultures for scalable cardiomyocyte derivation from iPSCs. Several bioreactor protocols have been developed over the past 10+ years by academic and industry labs. Output of 1.2 million/mL with >90% purity of for the most part terminally differentiated cardiomyocytes is rather stand. Bioreactor volume (100 mL) is pretty small and scalability to larger volumes (>1 L) would need to be demonstrated to make a stronger point.

Specific comments:

The authors should state more clearly why they believe that their process is an advance over the state-of-the-art.

Small bioreactors are easy to run with high output. Scalability to mass production is much more complicated. The authors should at least go to the 1 L level to demonstrate some evidence for robust and reproducible scalability.

The controlled rate freezing protocol (full temp ramp and holding steps) has to be provided in full detail.

Proliferation rate (19%) appears quite high. This does not argue for advanced maturation. In fact, all statements as to maturity differences must be deleted. The reported endpoints are insufficient to support a meaningful maturity claim.

Failure of mCM to respond to electrical stimulation suggests poor quality. This and the lower purity may have contributed to lower seeding (on patterns and in EHT).

Lower force and sarcomere density in mCM EHT may be because of an unfavorable CM:non-CM ratio. For a proper comparison of mCM and bCM contractility in EHT, cell composition must be controlled and similar.

Several iPSC lines are mentioned in the abstract and methods section. The presented data seems to be for the most part from the dox-inducible Cas9-inserted Coriell control line. It would be important to at least show data as to cardiomyocyte yield from the different iPSC lines to support the protocol robustness claim.

Response to Reviewers

GENERAL REMARKS

We would like to thank the reviewers and editorial team for their feedback on our manuscript. We extensively revised the manuscript to comprehensively address the comments. Some highlights of the main experiments and findings included in the revision to respond to the critiques:

- We performed additional cardiac differentiations in the bioreactor using a total of 14 hiPSC lines from 6 different donors resulting in 1.21 million/mL bioreactor-derived cardiomyocytes (bCMs) with ~94% purity (25 differentiations). These differentiations used exactly the same protocol (no adjustment of CHIR concentrations).
- We applied our optimized cardiac differentiation protocol to a spinner flask suspension system, which is far more available to academic labs than a bioreactor. We found that the spinner flasks yielded 1.79 million/mL hiPSC-CMs with ~94% purity (6 differentiations, 1 hiPSC line). Scaling of this protocol by 3.8-fold resulted in the production of 1320 million hiPSC-CMs in a single differentiation with 3.4 million hiPSC-CMs per mL, at present the highest concentration reported. We named these cells spinner flask-derived hiPSC-CMs (sCMs), and repeated the majority of experiments to enable comprehensive comparison to bCMs.
- We found conditions in which monolayer cardiomyocytes could be paced and then did an extensive set of experiments to compare fresh and cryo-recovered iPSC-CMs head-to-head. Interestingly, cryo-recovered bCMs displayed features of more mature Ca^{2+} transients and action potentials compared to all other conditions.
- Monolayer-differentiated hiPSC-CMs (mCMs) were treated with lactate (mCM+L) to assess the effect of metabolic maturation and compared to our experimental groups.
- We validated the metabolic maturation of bCMs, sCMs, and mCM+L by analyzing mitochondrial respiration in bCMs, mCMs+L, mCMs-L, and sCMs. bCMs and sCMs had comparable mitochondrial function to mCM+L. All three of these groups had significantly higher mitochondrial function compared to mCM-L. These functional measurements were consistent with single cell transcriptomics measurements, which demonstrated enrichment of mitochondrial function and electron transport chain in bCMs compared to mCMs.
- Finally, we report the first bioreactor-derived cardiac organoids (bCOs) fully generated in suspension, which was achieved by minor modifications of the bioreactor suspension culture protocol. In addition to a basic description of the bCO model, we also evaluated cellular composition by scRNAseq. Consistent with previously published cardiac organoid models, the majority of cells were hiPSC-CMs.

We summarized our main benchmarks for the cardiac differentiation protocol and functional results in Table S1, which also includes other published cardiac monolayer and suspension differentiation protocols to enable an easy comparison.

REVIEWER COMMENTS

Reviewer #1 (Remarks to the Author):

The paper by Prondzynski et al described a method which used a stirred suspension bioreactor to produce on average 124 million hiPSC-CMs with >90% purity using a variety of hiPSC lines (19 differentiations; 10 iPSC lines).

The novelty of the study is weak. The methods using bioreactors to differentiate human pluripotent stem cells not new, as several other groups have reported already (doi:10.1038/s41596-019-0189-8; doi.org/10.1172/jci.insight.99941; doi: 10.3389/fbioe.2021.674260). The only difference may be the CMs generated each batch is larger in the current paper as compared with others.

We disagree with the reviewer on this point. This manuscript offers novelty and fills in critical gaps in the literature in the following ways:

1. There have been other studies reporting production of iPSC-CMs from bioreactors. However, it is crucial to note that the iPSC-CMs generated in those prior studies lacked functional characterization. Therefore it is impossible to know about the quality or functional properties of the cells, and whether it is worthwhile to invest in the hardware and time to implement the protocols in the lab. In contrast, we extensively characterized the cells and showed that their properties after cryo-recovery are superior to cells derived by standard 2D monolayer differentiation. This fills in a critical knowledge gap that is critical for further applications of these cells.
2. In the revised manuscript, we extend the suspension culture protocol to include spinner flasks and extensively compare iPSC-CMs generated in bioreactors to spinner flasks. Although bioreactor cells were superior, particularly in force production, in many functional aspects spinner flasks were nearly as good and offered the advantages of far lower hardware cost and greater scalability.
3. In the revised manuscript, we added initial characterization of the first cardiac organoids generated entirely in suspension culture. This offers far greater scalability than current cardiac organoid protocols, in which one organoid is generated per well of multiwell dishes.

The reviewer cited several studies that also reported iPSC-CM maturation or differentiation protocols. In comparison to these studies, our manuscript makes significant advances:

1. The study by Ronaldson-Bouchard et al. (doi:10.1038/s41596-019-0189-8) focused on the development of an electromechanical maturation protocol for 3D-engineered heart tissues (EHTs) independent of cardiac differentiation method, in contrast to our study. Furthermore, EHTs were generated by mixing human dermal fibroblasts with hiPSC-CMs. However, cryopreserved bioreactor-derived hiPSC-CMs from our study reached comparable levels of contraction force as reported in Ronaldson-Bouchard et al. without electromechanical maturation or cell supplementation.

2. In the study by Cyganek et al. (doi.org/10.1172/jci.insight.99941), the authors developed a cardiac subtype-specific differentiation protocol for the generation of atrial hiPSC-CMs, which were experimentally compared to hiPSC-CMs generated without retinoic acid supplementation. Our study focused on the development of an optimized cardiac differentiation protocol in suspension and functional comparison of suspension-derived hiPSC-CMs to monolayer-derived hiPSC-CMs.
3. Our study references the report by Khan-Krell et al. ([doi: 10.3389/fbioe.2021.674260](https://doi.org/10.3389/fbioe.2021.674260)) in the main manuscript and we show reported differentiation benchmarks, functional experiments and resulting values side-by-side in Table S1. Khan-Krell et al. utilized a shaker flask suspension culture system in contrast to our study, which used a bioreactor and suspension flask system. As with many manuscripts reporting differentiation protocols, this study lacked functional characterization of the produced cells.

3D bioreactor promotes maturation of CMs is already known. Authors shall provide detailed analysis of structure, like EM, ion channel, electrophysiology, and metabolism to show that their method is better than others.

The reviewer suggested that the prior literature already demonstrated that 3D bioreactor promotes maturation of CMs, but does not provide supporting citations. In Supplementary Table 1, we summarized prior suspension culture iPSC-CM differentiation studies including their evaluation of CM function and maturity. There is one manuscript that previously reported metabolic maturation of CMs in 3D differentiation in bioreactors (PMID: 29178315), which is referenced in the main manuscript and compared to our study in Table S1.

Our conclusion that bioreactors promote CM maturation is based on the following evidence:

1. Morphological comparison of bCMs to mCMs, including measurements of circularity and sarcomere organization in patterned and unpatterned assays (Fig. 3 and Fig. S3).
2. scRNAseq analysis showed a higher portion of ventricular-like and overall CMs in bioreactors and higher expression of metabolic genes when compared to monolayer CMs. In the revised manuscript we analyzed differentially expressed genes between bCM and mCM cells within the principal cardiomyocyte clusters (0, 1, 2; Fig. 2e) and showed that genes more highly expressed in bCMs were strongly enriched for electron transport chain, mitochondrial respiratory chain complexes, and muscle contraction (Fig. 2f, left). In contrast, genes more highly expressed in mCMs were highly enriched for glycolysis, extracellular matrix organization, and heart development (Fig. 2f, right).
3. Comparison of calcium transients in bCMs, sCMs and mCMs \pm L from cryopreserved and fresh hiPSC-CMs (Fig. 3l-n and Fig. S6c-g) showed highest amplitudes and upstroke velocities in cryopreserved bCMs.
4. Comparison of voltage transients in bCMs, sCMs and mCMs \pm L from cryopreserved and fresh hiPSC-CMs (Fig. 3o-q and Fig. S6h-l) showed highest upstroke velocities in cryopreserved bCMs, a hallmark of ventricular function.
5. Analysis of mitochondrial function using the Seahorse mitochondrial stress test. This showed that suspension-derived, bCMs and sCMs, developed higher basal and maximum mitochondrial respiration rates and higher ATP production than mCMs prior to lactate treatment (Fig. 3r and

S6m-o). Furthermore, these levels of respiration and ATP production were comparable to lactate-treated mCMs generated in our study.

6. Analysis of bCM vs mCM contractile function, using engineered heart tissues (EHTs). This analysis showed considerably stronger contractile force, formation of homogenous tissues and increased sarcomere spacing in bCM EHTs. (Fig. 4).

One issue is the 10 hiPSC lines: as shown in Fig 1B, did the first 5 hiPSC lines originate from the same one: WTC-Cas9? If so, probably they share the similar genetic and epigenetic background even after CRISPR edited

In the revised manuscript, we have now included a broader experience with different iPSC lines, and we distinguished sublines originating from a shared parental line from independent cell lines, which is reflected in the main manuscript and in Figs. 1 and 5. To summarize, this paper includes WTC and 8 genome-edited derivatives; a PGP1 genome-edited derivative; and 4 patient-derived iPSC lines. This number of lines far exceeds prior publications on iPSC to cardiomyocyte differentiation protocols (see Table S1).

Reviewer #2 (Remarks to the Author):

The paper describes an optimized method for producing high numbers of hiPSC-cardiomyocytes at high purity. Whilst multiple methods have been described for this previously, many of these are used by companies and are not openly accessible to the research community. This method appears robust and is claimed to be reproducible across multiple (10) hiPSC lines and differentiations although not all claims are supported or documented by the data presented, particularly reproducibility between lines. However, the functional properties of the derivative hiPSC-CMs are at least equal to or better than those described for standard monolayer differentiation which is encouraging. There are a number of questions that need to be addressed if the protocol is to be of real value to the field.

Major points:

1. To what extent is the protocols dependent on this particular brand of bioreactor? It is widely known that bioreactors differ in their properties and in general each type needs optimising. The timings in the protocol, cell seeding densities etc may be dependent on the bioreactor properties. If it is not possible to test the protocol using another brand, then the authors could discuss how long it took to optimise this protocol, which parameters they changed etc. This would be of value to users starting with different bioreactor platforms.

We appreciate the rationale behind this question. However, this question is difficult for us to answer as each bioreactor system requires a major hardware investment (over \$150k), so testing across multiple

systems is not feasible. Moreover, functional testing of the cells produced by each system would be a major undertaking.

We did have access to an alternative system, which is a simple suspension culture system that does not continuously adjust pH, pO₂, or pCO₂. Success with this system would greatly democratize suspension culture, since the hardware cost is nominal (about \$5k). Therefore we tested the suspension culture protocol in spinner flasks and found that it generates equivalent numbers if not more iPSC-CMs using exactly the same differentiation protocol parameters, such as hiPSC inoculation density, culture volume, speed of rotation, CHIR and IWR concentrations and time intervals. Additionally, in many functional parameters, these cells were close to those generated in bioreactors and superior to monolayer culture with or without lactate enrichment. However, in EHT force generation, the spinner flask iPSC-CMs were closer to monolayer-derived cells and clearly lacked the exceptional force generation of bioreactor-derived iPSC-CMs. Furthermore, we showed that the spinner flask system scales well, allowing us to produce over 1E9 iPSC-CMs in a single 380 ml culture. A comprehensive section on spinner flask differentiation and functional characterization was implemented into the main manuscript (Fig. 5 and Fig. S9).

2. It would be helpful to show in each of the legends which hiPSC line is exactly being used. Most seem to have been done with the WCT-Cas9 line with or without the TNNI1-GFP reporter. For example, in the first section of the results “Optimized bioreactor differentiation protocol” there seems to be no reference to the other 10 lines tested. The same is true for the others sections: all seem to have been carried out using the WCT line.

In the revised manuscript, we improved labeling of hiPSC parental lines, sublines (i.e. products of genome editing), different experimental groups and included more legends in the figures to minimize confusion. Functional characterizations were solely done with the WTC-Cas9 line (see methods), which was differentiated either in bioreactor, spinner or monolayer cultures with consistent color coding throughout the manuscript (Figs. 2-5 and Figs. S.3-10). Additionally, we specify in the main manuscript that the optimized bioreactor protocol was tested in a variety of cell lines. However, functional analysis was just performed for WTC-Cas9 control line: “*For functional comparison to bCMs, we used healthy control iPSCs (WTC-Cas9) and differentiated them in parallel in adherent monolayers using the same protocol ... (Fig. S1a’).*” Functional comparison of all cell lines generated with our optimized differentiation protocol exceeds the scope of this study.

3. Could the authors explain the higher degree of maturity in the bCMs versus the mCMs. Was this related to the other (non-cardiomyocyte) cell types present? Eg endothelial cells versus fibroblasts?

Based on our data and previous studies it seems to be a mix of several factors that explains a higher degree of maturity in suspension-derived hiPSC-CMs:

- It has been previously shown that analogous surface proteins will adhere easier and faster in 3D aggregates (PMID: 26658353). This principle was tested for cardiac suspension cultures by Nguyen et al. (PMID: 25254340). The authors were able to show that heterogeneous populations

containing ~10%–40% cardiomyocytes, were able to purify in 3D aggregates to ~80%–100% cardiomyocytes, which corresponded to an enrichment factor of up to 7-fold.

- Next to homogenous adhesion protein profiles, it also has been shown that 3D aggregates promote structural and functional maturation of CMs due to their ability to better mimic the in vivo environment and to facilitate cellular communication via a variety of paracrine, autocrine and endocrine factors (PMID: 25254340; PMID: 20001738; PMID: 22662278, see supplementary table S1).
- Finally, previous comparison of chondrogenesis in static and perfused bioreactor culture (PMID: 11027186) showed the importance of stable pH as cells derived from bioreactors showed increased cell proliferation in comparison to static cultures, in which pH decreased much more rapidly and was less stable when compared to bioreactors.
- In our study both bCMs and sCMs had improved functional properties compared to mCMs, in line with the findings described above. However, the importance of continuous control of pH, pO₂, and pCO₂ was most clearly demonstrated by the superior force generated by bCM EHTs compared to sCM EHTs (or mCM EHTs).
- Previous studies that supplementation of non-CMs in 3D-engineered heart constructs (PMID: 29618819) or optimization of a differentiation medium that is supportive of a wider specification and differentiation of non-CMs and CMs simultaneously (PMID: 37105170) improved cardiomyocyte function. Quantities of non-CMs cells, determined by scRNAseq, were very low in the bioreactor cultures. Additionally, we did not supplement 25% of human dermal fibroblasts to our EHT constructs, yet we detected comparable forces as previously reported for fibroblast-supplemented EHTs (PMID: 29618819). Furthermore, we did not see the formation of an endothelial vascular-like network as reported previously (PMID: 37105170) in our bCM EHTs or bioreactor-derived cardiac organoids (bCOs). Therefore, we think that the main effects of maturation are attributed to the controlled environment and the benefits of suspension aggregate culture as described above.

4. Regarding the inability of mCMs to follow pacing at 1Hz: this seems somewhat strange as many reports show that pacing is possible. Do the authors have an explanation?

This observation was also somewhat surprising to us. In order to improve this portion of the manuscript, we tested pacing capabilities of mCMs 7 days after replating and found that they could be paced after cryo-recovery. In the revised manuscript, we therefore included an extensive comparison of all iPSC-CMs generated in this study (bCMs, sCMs, mCMs-Lactate and mCMs+Lactate) for “cryo” vs. “fresh” conditions 7 days after replating (Figs. 3 and S6; Figs. 5 and S9).

5. Regarding sarcomere length in mCMs versus bCMs, the difference between 1.6um and 1.7um (fig4j) seems very small to claim they were different: was it statistically significant over many EHTs measured at multiple positions in the EHTs? Eg were the measurements made in the middle or near the poles of the EHTs? Examining 3 and 2 EHTs respectively seems on the low side.

In the revised manuscript, we examined in total 367 (bCM) and 366 (mCM) sarcomere intervals for 66 (bCM) or 59 (mCM) regions of interest in 4 (bCM) or 5 (mCM) EHTs from two independent differentiations. This sample size is comparable to a previous report of cardiac sarcomere length measurements in EHTs (PMID: 32697997: n = 100 sarcomere intervals, 2 independent differentiations). With this expanded sample size, we observed that mean sarcomere length was 1.75 μm and 1.64 μm for bCM and mCM EHTs, respectively. This difference was highly significant ($P < 0.0001$). The EHTs were sampled from a consistent location near the middle of the muscle bundles (shown below). This difference is consistent with other measurements showing that bCMs have greater overall maturity, which collectively resulted in bCM EHTs producing far more force than mCM EHTs.

6. Loss of functional properties after cryopreservation (ref #9): is this the case here? Not all studies have reported cryopreservation is detrimental to hiPSC-CMs (e.g., DOI: 10.1016/j.scr.2019.101698). This claim needs to be supported by head-to-head comparisons of single parameters on the same CM batch before and after cryopreservation. Which functional parameters changed and which did not?

We agree with the reviewer that this is an important question. Therefore, we included extensive functional characterization of cryorecovered and fresh bCMs, sCMs and mCMs (+ and - lactate treatment) in the revised manuscript (Figs. 3 and S6; Figs. 5 and S9). Surprisingly, we found that in head-to-head comparisons in many cases cryo-recovered bCMs and sCMs had greater functional properties than fresh cells. This is in line with the study mentioned by the reviewer (DOI: 10.1016/j.scr.2019.101698) as the authors found that cryopreservation is not detrimental to function and can even promote the maturation of hiPSC-CMs to a ventricular subtype, which was clearly reflected by cryo-recovered suspension-derived iPSC-CM voltage transients (Figs. 3o and 5r). We've added this reference to the discussion. Additionally, our results for cryo-recovered suspension-derived iPSC-CMs contrasts with the report of Zhang et al. (PMID 33338435) showing loss of some functional characteristics after cryo-preservation. This study reported ~60% cell viability after cryo-recovery, whereas our viability was ~94%. Therefore it is likely that the cells in Zhang et al. experienced considerably more trauma, which negatively impacted their physiological properties.

Minor points:

1. Abstract: hiPSC-CMs have not actually yet proved to be valuable for cardiac regeneration. This is not then limiting clinical translation in regenerative medicine at least. Please re-word.

We rephrased the sentence: "In the last decade human iPSC-derived cardiomyocytes (hiPSC-CMs) have been used for cardiac disease modeling and studies of cardiac regeneration, yet challenges with

scale, quality, inter-batch consistency, and cryopreservation remain, reducing experimental reproducibility and limiting clinical translation.”

2. Please be consistent in use of hiPSC-CM versus iPSC-CM.

We adopted iPSC-CM throughout.

Reviewer #3 (Remarks to the Author):

The manuscript by Prondzynski and colleagues describes a bioreactor protocol for the generation of cardiomyocytes from human iPSCs. They compare the bioreactor CMs (bCMs) to CMs they generate by monolayer differentiation using a protocol based on the bioreactor conditions. They claim that it is more efficient to generate bCMs and that these bCMs are more mature than monolayer CMs.

As the authors acknowledge there have been a number of publications over the last few years describing the generation of CMs using bioreactors, which in the authors’ opinion these have been plagued by issues in terms of inconsistent differentiations. Although they claim to have addressed this reproducibility issue by generating bCMs from 10 different iPSC lines, from the data provided it is difficult to be convinced of this.

As referenced by the reviewer there have been other studies reporting production of iPSC-CMs from bioreactors. However, earlier manuscripts reported suspension culture protocols, but the lack of functional studies of the resulting iPSC-CMs made it difficult for research groups to decide whether this approach was worth the investment in equipment and expertise. We report a bioreactor protocol for producing iPSC-CMs with confirmed robust functional properties and improved consistency compared to standard monolayer differentiation protocols. Furthermore, we give detailed descriptions of differentiation benchmarks and freeze/thaw procedures that increase reproducibility between batches. Overall, we achieve improved consistency compared to current commonly used small batch monolayer differentiation protocols by (1) producing large batches of cells that can be cryo-recovered and so used in many experiments over time; (2) increasing the consistency between batches.

Additionally, we extended the suspension culture protocol to include spinner flasks and extensively compare iPSC-CMs generated in bioreactors to spinner flasks. We successfully applied and scaled the same cardiac differentiation protocol to spinner flasks yielding even higher iPSC-CMs numbers than bCMs differentiations (Fig. 5). This greatly democratizes suspension culture, since the hardware cost for spinner flasks is nominal in comparison to bioreactors (about \$5k vs \$150k, respectively). This might enable more labs worldwide to generate suspension-derived iPSC-CMs using this low-cost protocol (see supplementary table S3, *Bioreactor versus Monolayer Costs*), while counteracting the worldwide reproducibility crisis in the life sciences.

Finally, we have now included a broader experience with different iPSC lines, and we distinguished sublines originating from a shared parental line from independent cell lines, which is reflected in the main

manuscript and in Figs. 1 and 5. To summarize, this paper includes WTC and 8 genome-edited derivatives; a PGP1 genome-edited derivative; and 4 patient-derived iPSC lines. This number of lines far exceeds prior publications on iPSC to cardiomyocyte differentiation protocols (see supplementary table S1). However, functional characterizations were solely done with the WTC-Cas9 line (see methods), which was differentiated either in bioreactor, spinner or monolayer cultures (Figs. 2-5 and Figs. S.3-10). Functional comparison of all cell lines generated with our optimized differentiation protocol exceeds the scope of this study.

How many differentiations were performed in total per line? For several of the lines it appears that only 1 differentiation was performed, making it difficult to evaluate the reproducibility. Indeed for patient#4 it does not appear that they managed to successfully differentiate it. Additionally 50% of the lines are sub clones from the same parental line (WTC) - these cannot be considered completely independent lines. It is also not clear how the MCBs used by the authors differ compared to how researchers usually stockpile and characterise their undifferentiated cells. More details regarding this need to be provided.

As mentioned in the previous answer, we now included a broader experience with different iPSC lines, and we distinguished sublines originating from a shared parental line from independent cell lines, which is reflected in the main manuscript and in Figs. 1 and 5. To summarize, this paper includes WTC and 8 genome-edited derivatives; a PGP1 genome-edited derivative; and 4 patient-derived iPSC lines. This number of lines far exceeds prior publications on iPSC to cardiomyocyte differentiation protocols (see supplementary table S1). However, we preferred to show broad applicability of the optimized cardiac differentiation protocol to different cell lines so 8 lines were just differentiated once. Nevertheless, the control line (WTC-Cas9) has been differentiated several times in the bioreactor (8x) and in spinner flasks (6x). The patient#4 line mentioned by the reviewer can indeed be found in the “failed” column (Fig. 1b). However, it can also be seen in the >90% TNNT2⁺ column. We show one example for a successful and for a failed differentiation using the patient#4 line.

In contrast to the bioreactor protocol validation, functional characterizations were solely done with the WTC-Cas9 line (see methods), which was differentiated either in bioreactor, spinner or monolayer cultures (Figs. 2-5 and Figs. S.3-10). For detailed analysis of batch-to-batch variability and comparison between protocols, we focused on one iPSC line and compared bCMs vs mCM in three separate differentiations using each method. Functional comparison of all cell lines generated with our optimized differentiation protocol exceeds the scope of this study.

While some labs use master cell banks to increase consistency between batches, this is not a uniform practice. Here we describe the use of MCBs to enhance its adoption as best practice and to indicate that it likely contributes to minimizing batch-to-batch variation. In fact, none of the cardiac differentiation protocols referenced in this study mentioned the use of MCBs (see Table S1).

The authors also link the SSEA4% and size of the EBs to whether a bioreactor run is likely to be successful, but it is not obvious how predictive this is as one-third of the failed runs had an SSEA4 % above the 70% cutoff. Likewise, while they claim that EB diameter should be between 100-300um, more than half of the failed runs had EBs within that range.

Indeed, we show SSEA4 and EB diameters in the manuscript and claim them as critical benchmarks for successful differentiation (Fig. 1b,c). Both panels consist of the same runs for failed and successful differentiations (>90% TNNT2⁺), therefore single differentiations can be followed and evaluated on both parameters. As the reviewer correctly pointed out there are differentiation runs that failed, but show high SSEA4 values. Failed runs with high SSEA4 values showed EB diameter out of optimal range and therefore failed. Vice versa, Failed runs with low SSEA4 values (<70%) showed optimal EB diameter. This is very well exemplified by above-mentioned patient#4 line. This line had optimal EB diameter (Fig. 1c, Failed column), but low SSEA4 (66%; Fig. 1b) and therefore yielded iPSC-CMs with low purities (Failed). Since differentiations were successful for hiPSC with SSEA4 value >70% (e.g. 71% and 78%), we decided to set the threshold for successful differentiation >70% SSEA4 (Fig. 1b). Similarly, all differentiation runs that were >70% SSEA4 and had an EB diameter within 100-300 μm were successful (>90% TNNT2⁺). To directly correlate SSEA4 and EB size please see the graph below showing a subset of successful (green, n=16) and failed runs (red, n=8). Some differentiations did not have both measurements, so this graph lacks some of the data points compared to the separate graphs in the manuscript.

Likewise for the comparison to the monolayer differentiation, it is not clear how frequently this was performed and appears to have only been done on the one cell line that failed the most in the bioreactor differentiations. Therefore when comparing yield for example (fig 1e) it should not be compared to other independent cell lines which clearly were the better performing ones in the bioreactor. It is also not obvious what steps the authors undertook to optimise the monolayer differentiation conditions. Seeding density is known to be a key factor in influencing the differentiation efficiency. As well it cannot be assumed that the same concentration of the small molecules used for the bioreactor differentiations will be optimal for the monolayer differentiations.

We used our standard iPSC line, WTCas9, derived from WTC11, for the bCM to mCM comparison. This cell line differentiates robustly into iPSC-CMs and is widely used by other labs (PMID: 33951429; PMID: 34019794; PMID: 37416454). The reason it appeared to have failed in some bioreactor differentiations is that we included our early runs that overlapped our learning curve and were lower in yield. The same can be observed for the newly added spinner flask differentiations (Fig. 5d). However, low

yield of iPSC-CMs in suspension cultures did not affect high-quality CM properties. Our lab routinely uses WTCCas9 for monolayer differentiation (e.g., PMID: 37595583) and we have extensively optimized conditions (cell density, CHIR concentration, IWR concentration) for the differentiation of this cell line. We tried to perform exactly the same bioreactor differentiation protocol in monolayers and therefore kept CHIR concentrations at 7 μ M for all differentiations, even though it is known that this value can range. However, one unique aspect of our bioreactor protocol is the fact that CHIR concentrations do not need to be adjusted for individual lines. On the contrary we had to adjust CHIR incubation time from 24h to 48h in monolayer differentiations (Fig. S1a,a') due to consistently failed differentiations with 24h CHIR. To further strengthen the bioreactor protocol we did not perform metabolic selection or MACS sorting in bioreactor or monolayer differentiations. Usually, this is done in the lab and we added several sections to the revised manuscript functionally comparing the suspension culture cells to mCMs with and without lactate treatment (Figs. 3 and S6; Fig. S8). We did not perform MACS purification (Miltenyi, PSC-Derived Cardiomyocyte Isolation Kit, 130-110-188) of monolayer generated iPSC-CMs in this study. Although this could increase the amount of TNNT2⁺ cells compared to bCMs.

Similarly the conclusions made from the qPCR and scRNAseq are questionable. The larger inter-batch variation in ACTN2 levels seen in the mCMs is due to this gene not detected at all in one batch of mCMs - indicating that either the authors are including failed monolayer differentiations which produced no CMs in their samples (which should then also apply to the bCMs), or there was a technical issue with those measurements.

We believe that the reviewer is commenting on Fig. 1i,i'' in the revised manuscript, in which we compare ACTN2 mRNA levels from bCMs and mCMs at day 15 (Fig. 1i) and show individual differentiation batches of mCM ACTN2 expression side-by-side (Fig. 1i'). Sampling for molecular analysis was randomized for each monolayer batch at day 15 of differentiation before final dissociation. Therefore, it could have led to taking samples of a monolayer well with very low cardiomyocyte %, which is most likely what happened here. These data and the newly added movie S4, which shows variable differentiation outcomes in each well of a 12-well plate, represent the reality of well-to-well variation faced by many users of current monolayer protocols. These failed wells were excluded for all other analyses. For all other analyses, we pooled mCMs from pre-selected visually beating wells and excluded wells that had no or reduced beating areas. As a result, these pooled mCMs had at least 60% TNNT2⁺ cells (Fig. 1d,e). Our scRNAseq data was also generated with pooled mCMs from the same batch and therefore pre-selected for beating and did not include wells with inefficient differentiation to beating CMs. Reproducibility of this analysis can be seen in Fig. S2a, in which we show biological replicates of scRNAseq analysis side-by-side, which are in agreement between biological duplicates (Fig. S2a).

The authors also highlight the higher expression of COL1A1 and COL3A1 in the mCMs at d5-7, but do not comment that by d8 expression levels of these genes are either the same or lower than what is observed in the bCMs, questioning whether this does reflect a greater proportion of non-CMs.

We felt that the cell composition of cultures was better analyzed by the scRNAseq data presented in Fig. 2, so we removed these panels from the revised manuscript.

The authors also note that the differentiation appears to be delayed in the monolayer vs bioreactor differentiations. It is also surprising that from the scRNAseq only 50% of the monolayer-differentiated cells are classified as CMs, despite cTnT FACS indicating the purity to be ~75-80% based on fig 1f. How do the authors explain this especially when the correlation between cTnT and scRNAseq-predicted CM content for the bCMs is much similar.

As correctly observed by the reviewer, % of TNNT2⁺ cells assessed by FACS did not overlap fully with % of CMs found in the scRNAseq data. Differentiation batches that were used for scRNAseq analysis had the following % TNNT2⁺ cells measured by FACS: bCMs (94% and 90%) and mCMs (64% and 70%). One reason for observed differences might be more stringent clustering of cell categories by scRNAseq. As can be seen in the feature plots below *TNNT2* gene expression is widely seen in CM clusters 0, 1, 2, 4, 5, 8, 9. However, *TNNT2* expression can also be seen in clusters 6 and 7 assigned to skeletal and smooth muscle, respectively, which account for ~23% of all cells in mCMs. In FACS analysis these cells show most likely as TNNT2⁺ cells even though they are not characterized as CMs in the scRNAseq, but still express TNNT2 protein. In conclusion, scRNAseq has a higher resolution for identifying different cell types based on hundreds of expressed genes, whereas FACS analysis is restricted to one protein. Therefore, FACS analysis has a much lower specificity and can also include non-CMs, that are either in the process of differentiating to CMs, or are other muscle cells that express TNNT2, especially at this early stage of cardiac specification.

The authors also state that they have managed to solve various issues surrounding the cryopreservation of hPSC-CMs, in particular those surrounding the percentage of CMs that are recovered and functional impact that freezing might have on the CMs. Whether cryopreservation does in fact have a negative functional impact remains debatable. The authors have referenced an article that did observe this, however other publications (e.g. Miller et al *Curr Protoc Stem Cell Biol*; van den Brink et al *Stem Cell Res*) do not observe such issues. Likewise many companies are providing cryopreserved hPSC-CMs, which can be paced. The percentage of CMs recovered that the authors report (>90%) is much higher than what others have observed, but how this was determined and at which point in the thawing procedure is not described in the paper. Likewise details on the optimisations they performed

related to cell dissociate and cryo-protectant media are not described. The freezing density also varies greatly (between 1×10^6 - 20×10^6 /ml). Did this have any influence on recovery or functionality?

Miller et al. reported a protocol for cryo-recovery of iPSC-CMs. They show that the cells have properties of cardiomyocytes after cryo-recovery (activity on MEA, expression of cardiomyocyte genes) but there is no comparison before and after cryopreservation, and there is no data on pacing. Additionally, their section on “Thawing and Resuspension” is described to the same depth as in our method section. Van den Brink et al. reported that iPSC-CMs can be successfully cryopreserved and recovered. They compared action potential parameters and qualitative cell motion during 1 Hz pacing. These cells were shown to respond to 1 Hz pacing after cryo-recovery. It should also be pointed out that the conditions for pacing are very important, e.g. confluent monolayers as described in the van den Brink manuscript are more easily paced than sparse cell islands. Nevertheless, these manuscripts (and the availability of commercial cells, although methods for production and preservation of those cells are not publicly available) suggest that some differentiation-cryopreservation pipelines have at least some degree of success.

In our hands, the same iPSCs differentiated by mCM protocol lost monolayer paceability at 4 days after cryo-recovery. However, in the revised manuscript we show that these cells were paceable by 7 days after cryo-recovery. bCMs were paceable at both time points. Using the 7 day post-cryorecovery time point, in the revised manuscript we extensively compared the physiological properties of cryo-recovered and fresh bCMs, newly added sCMs and mCMs (\pm Lactate; Figs. 3 and S6; Figs. 5 and S9). We also provide detailed methods for cryo-preservation and thawing to achieve high viability, which was determined by Trypan blue staining and can be found in the method section of the manuscript. Interestingly, van den Brink et al., showed that cryopreservation can even promote the maturation of hiPSC-CMs to a ventricular subtype, which was clearly reflected by cryo-recovered suspension-derived iPSC-CM voltage transients in our revised manuscript (Figs. 3o and 5r). This reference was added to the main manuscript.

Additionally, our results for cryo-recovered suspension-derived iPSC-CMs contrasts with the report of Zhang et al. (PMID 33338435) showing loss of some functional characteristics after cryo-preservation. This study reported $\sim 60\%$ cell viability after cryo-recovery, whereas our viability was $\sim 94\%$. Therefore it is likely that the cells in Zhang et al. experienced considerably more trauma, since cells were dissociated with Accutase, no defined freezing media (Serum-based) and isopropanol-filled containers were used, which likely negatively impacted their physiological properties. Chen et al. (PMID: 26318718) reported a viability of $\sim 85\%$, which is significantly improved to Zhang et al., and most likely attributed to the usage of commercial freezing medium and a controlled rate freezer.

Cell density did not affect viability during cryo-preservation and recovery, as we tested 1, 2, 5, 10 and 20 million for bCMs and 2 and 80 million aliquots for sCMs resulting in high viabilities ($>90\%$).

Lastly, they also observe a higher percentage of multi-nucleated mCMs compared to bCMs, and comment that multinucleation tends to indicate cardiac disease. This % of multi-nucleated CMs, even for the bCMs is much higher than what is observed in other publications. Indeed in the Mosquera paper that they cite, for the disease CMs these values were less than 2%. Similarly the authors believe the mCMs show a higher degree of DNA damage following thawing. What do they think the reason for this

is, and have they ruled out that this is not due to the cell line used for the differentiations modified to be able to express Cas9. Again use of the original WTC1 cell line here would have been better.

We disagree with the reviewer on the point of reporting higher multinucleation rates than previously reported for hiPSC-CMs (PMID: 2974161; PMID: 36078153). Mosquera et al. (PMID: 2974161) reports ratios of ~80% mono-, 15% bi-, and 5% multinucleated WT hiPSC-CMs 7 days after replating (Fig. 3c,d Mosqueira et al.), generated from a healthy donor line (AT1). Our own results 7 days after replating show in fact lower multinucleation rates for bCMs, as we report for unpatterned sparse single cells ratios of ~88.8% mono-, 10.7% bi-, and 0.6% multinucleated bCMs and ~82.2% mono-, 14.4% bi-, and 3.4% multinucleated mCMs (Fig. 3d). Therefore our mCM data almost exactly reproduces what was previously published by Mosqueira et al using similar time frames, unpatterned substrates and different cell lines. Another study by Zech et al. (PMID: 36078153) evaluated multinucleation of hiPSC-CMs 30 days after replating resulting in ratios of ~80% mono-, 17% bi-, and 3% multinucleated WT hiPSC-CMs derived from an inhouse iPSC control line. Therefore, our own data is in line with previous studies using different donor control lines supporting no side effects caused by stably integrated Cas9 in our control iPSC line.

Furthermore, we believe that our patterned microsubstrates can be seen as a maturation platform for single hiPSC-CMs, which was confirmed by more mature sarcomere structure and nucleation ratios of ~78.7% mono-, 19.5% bi-, and 1.8% multinucleated in bCMs. However, mCMs showed less attachment and survival in these patterns, which was reflected by higher multinucleation rates (~66.7% mono-, 25.9% bi-, and 7.4% multinucleated mCMs; Fig. 1i) and genotoxic stress as indicated by increased DNA double-strand breaks (Fig. 3k).

Minor comments:

The authors should provide more details regarding the bioreactor setup they established. What was the rationale of using Rushton-type impellers (typical use for fermentations)? How was pH controlled and adjusted? What were the target, upper and lower limits? They also need to provide more details regarding the media change at day 3. They mention high O₂ consumption, but how was this determined by medium discolouration. It is not clear what the parameters for performing medium exchanges were.

We thank the reviewer for making us aware of our mistake concerning the Rushton-type impeller, which we now replaced in the method section with the 8-blade impeller (60° pitch) that was used in this study. Additionally, we added a more detailed description of pH, which was set at 7, and we defined high oxygen consumption as >30% XO₂.PV (Readout for Oxygen in the bioreactor system). Finally, we refer to the Nature protocols publication by Kempf et al., in the main manuscript and methods part, which used exactly the same Eppendorf bioreactor system and provides a detailed step-by-step preparation for running and calibrating technical aspects of this system (PMID: 26270394).

Table S1 - data regarding the ventricular CMs appears to be missing.

Table S1 now includes measurements of % CMs and % ventricular CMs.

Fig 1a - maintenance is spelt incorrectly

Corrected.

Reviewer #4 (Remarks to the Author):

Prondzynski and colleagues present a study on stirred bioreactor cultures for scalable cardiomyocyte derivation from iPSCs. Several bioreactor protocols have been developed over the past 10+ years by academic and industry labs. Output of 1.2 million/mL with >90% purity of for the most part terminally differentiated cardiomyocytes is rather stand. Bioreactor volume (100 mL) is pretty small and scalability to larger volumes (>1 L) would need to be demonstrated to make a stronger point.

We report a bioreactor protocol for producing iPSC-CMs with confirmed robust functional properties and improved consistency compared to standard monolayer differentiation protocols. Additionally, we describe handling of iPSCs, dissociation, freezing and thawing of hiPSC-CMs to provide a holistic framework for cardiac differentiation and in vitro modeling. Earlier manuscripts reported suspension culture protocols, but the lack of functional studies of the resulting iPSC-CMs made it difficult for research groups to decide whether this approach was worth the investment in equipment and expertise (Table S1).

We successfully increased the culture volume to 250 and 380 ml, with an increased yield of iPSC-CMs/ml at the higher volumes. In fact, 380 ml cultures (scaling by 3.8-fold) resulted in the production of 1320 million hiPSC-CMs in a single differentiation with 3.4 million hiPSC-CMs per mL, at present the highest concentration reported. We also extensively characterized the functional properties of the resulting iPSC-CMs and named them spinner-derived hiPSC-CMs (sCMs; Figs. 5 and S9). These additions to the manuscript illustrate the scalability of the suspension culture protocol and make it much more widely available to academic labs, many of whom could easily afford the spinner flask system but not the bioreactor.

Specific comments:

The authors should state more clearly why they believe that their process is an advance over the state-of-the-art.

We are not able to compare the functional properties of the cells produced by our protocol to those produced by other protocols – because no other study of suspension culture iPSC-CMs has reported the functional properties of the produced cells (Table S1). Some aspects of the cells produced by our protocol that differ from prior reports are (1) the appearance of functional beating as early as day 5; appearance of beating at day 7 is more typical (Table S1); We added the scRNAseq enrichment analysis (2) within the principal cardiomyocyte clusters (0, 1, 2; Fig. 2e) showed that genes more highly expressed in bCMs were strongly enriched for electron transport chain, mitochondrial respiratory chain complexes, and muscle contraction (Fig. 2f, left). In contrast, genes more highly expressed in mCMs were highly enriched for glycolysis, extracellular matrix organization, and heart development (Fig. 2f, right); metabolic maturation

was further supported by Seahorse analysis (3) revealing increased basal and maximum mitochondrial respiration, and higher ATP production comparable to mCMs treated with lactate; Rod-shaped morphology in unpatterned substrates (4) and higher sarcomere regularity (5) in patterned bCMs; Comparison of calcium transients (6) in bCMs, sCMs and mCMs±L from cryopreserved and fresh hiPSC-CMs showed highest amplitudes and upstroke velocities in cryopreserved bCMs; Comparison of voltage transients (7) in bCMs, sCMs and mCMs±L from cryopreserved and fresh hiPSC-CMs showed highest upstroke velocities in cryopreserved bCMs, a hallmark of ventricular function; and (8) the high force generated by bCM EHTs. We used the same EHT setup as that originally developed by the Eschenhagen group and commercialized through EHT Technologies now acquired by DiNAQOR. A comparison of force generated by our cells compared to all other publications using this system (and our mCM cells) showed that the bCMs generate ~2.5 times more force than prior studies and comparable forces generated as EHTs that underwent an electromechanical maturation protocol (PMID: 28492526; PMID: 31680489; PMID: 29618819).

Small bioreactors are easy to run with high output. Scalability to mass production is much more complicated. The authors should at least go to the 1 L level to demonstrate some evidence for robust and reproducible scalability.

As mentioned above, we are unable to perform these experiments given the limitations of the DASBox bioreactor system. Furthermore, we do not believe that most academic labs would be interested in this scale for the goal of disease modeling and increased reproducibility of iPSC-CM cultures. However, we did successfully scale the spinner flask to 380 ml and showed more than linear increase in iPSC-CM output within this range of scaling. Interestingly, Chen et al. (PMID: 26318718) performed 1 l scaling experiments resulting in 1.4 million hiPSC-CMs per mL yielding a total amount of 1400 million hiPSC-CMs. Our scaling approach in 380 mL generated 3.4 million hiPSC-CMs per mL yielding 1320 million cells in total. Therefore, we were able to generate the same amount of hiPSC-CMs in less than half of the total volume reported by Chen et al., further supporting applicability and scalability of our optimized suspension differentiation protocol in comparison to previous studies.

The controlled rate freezing protocol (full temp ramp and holding steps) has to be provided in full detail.

We provided details on temperature settings, ramps and holding steps in Table S7.

Proliferation rate (19%) appears quite high. This does not argue for advanced maturation. In fact, all statements as to maturity differences must be deleted. The reported endpoints are insufficient to support a meaningful maturity claim.

We have toned down statements regarding general cellular maturity, and we have further strengthened the data supporting the increases in specific morphological and functional properties (as described above). Additionally, transcriptome analysis of genes differentially expressed between bCMs and mCMs showed enrichment of terms related to mitochondria, oxidative metabolism, and heart

contraction in bCM upregulated genes, and terms related to glycolysis and heart development in mCM upregulated genes. These analyses were added to Fig. 2.

Regarding cell cycle, the single cell transcriptomics data obtained at Day 15 suggest that ~16% of bCMs and ~8% of mCMs are cycling. We are not convinced about the accuracy of cell cycle measurements by scRNAseq transcriptome analysis, therefore removed these analyses from the revised manuscript.

Failure of mCM to respond to electrical stimulation suggests poor quality. This and the lower purity may have contributed to lower seeding (on patterns and in EHT).

We do not believe that the differences we observed between mCM and bCM can be attributed to low quality mCM: 1. In standard EHT constructs, the mCM EHTs performed comparably to EHTs reported in the literature. This suggests that our mCMs are representative of what other investigators are currently using. 2. For the revised manuscript, we tested different cryo-recovery protocols to obtain paceable mCMs. We found that the mCMs were paceable at 7 days of culture after cryo-recovery, whereas they were not at 4 days of culture. This allowed us to compare the physiological properties of mCMs and bCMs in the revised manuscript. Moreover it shows that the mCMs are not intrinsically low quality. 3. Ratios of multinucleation were comparable in mCMs to other WT hiPSC-CMs previously studied in unpatterned substrates (PMID: 2974161; PMID: 36078153).

Lower force and sarcomere density in mCM EHT may be because of an unfavorable CM:non-CM ratio. For a proper comparison of mCM and bCM contractility in EHT, cell composition must be controlled and similar.

The force produced by bCM EHTs was much higher than that observed in other EHTs. This is likely a reflection of the intrinsic properties of bCMs rather than a difference in CM:non-CM ratio: For instance, bCMs and sCMs had similar CM:non-CM ratio, but bCMs produced much higher force. sCMs had higher CM:non-CM ratio than mCMs, but produced comparable force. We did make efforts to increase cardiomyocyte fraction in mCM cultures by lactate treatment, but we did not observe an increase in TNNT2⁺ cell percentage. Although mCM+L and mCM-L EHTs produced comparable force, these iPSC-CMs differed in other physiological properties (Ca²⁺ handling; AP; mitochondrial function), indicating that lactate treatment alters the intrinsic properties of iPSC-CMs.

Several iPSC lines are mentioned in the abstract and methods section. The presented data seems to be for the most part from the dox-inducible Cas9-inserted Coriell control line. It would be important to at least show data as to cardiomyocyte yield from the different iPSC lines to support the protocol robustness claim.

We now included a broader experience with different iPSC lines, and we distinguished sublines originating from a shared parental line from independent cell lines, which is reflected in the main manuscript and in Figs. 1 and 5. To summarize, this paper includes WTC and 8 genome-edited derivatives; a PGP1 genome-edited derivative; and 4 patient-derived iPSC lines (Fig. 1b-e). This number of lines far

exceeds prior publications on iPSC to cardiomyocyte differentiation protocols (see supplementary table S1). Additionally, we improved labeling of iPSC parental lines, sublines (i.e. products of genome editing), different experimental groups and included more legends in the figures to minimize confusion. Functional characterizations were solely done with the WTC-Cas9 line (see methods), which was differentiated either in bioreactor, spinner or monolayer cultures with consistent color coding throughout the manuscript (Figs. 2-5 and Figs. S.3-10). Additionally, we specify in the main manuscript that the optimized bioreactor protocol was tested in a variety of cell lines. However, functional analysis was just performed for WTC-Cas9 control line.

REVIEWER COMMENTS

Reviewer #1 (Remarks to the Author):

The paper introduces an optimized method for producing a large quantity of high-purity hiPSC-induced cardiomyocytes. While various methods have been previously discussed, this approach seems reliable and applicable across multiple hiPSC lines, yielding at least equivalent or superior functional characteristics in derived hiPSC-CMs compared to standard monolayer differentiation. This is encouraging, but there are still some issues to address.

Main concerns:

1. I remain confused about the author's response to the issue of low innovativeness in the rebuttal letter. I am not particularly clear on whether the author intends to convey the enhancement of cardiomyocyte differentiation efficiency and quality by the bioreactor or the subsequent rotating flask suspension system. The author appears to emphasize both products, each with its advantages and disadvantages.

The author acknowledges that the promotion of cardiomyocyte differentiation by the bioreactor has been reported before. Therefore, the novelty of this study, as suggested by the author, lies in the detailed functional identification of the generated cardiomyocytes, which, according to the author, was not meticulous in previous research. However, I believe this is not the innovation point. The highlight of this paper lies in achieving similar effects as adding human dermal fibroblasts with the addition of very few growth factors. The author could emphasize this point in the manuscript and clarify the improvements made with the bioreactor, which are not mentioned in the text.

In the subsequent discussion of the rotating flask suspension system, the paper mentions that although it is not as perfect as the bioreactor, it can achieve higher throughput of cardiomyocytes. In fact, the rotating flask suspension system has long been proposed (such as PMID: 26318718). Therefore, I hope the author can reorganize the writing to clearly state which system they believe is better for obtaining highly efficient and homogeneous cardiomyocytes and which is more suitable for widespread application. Once you have determined which one, the functional identification of the cardiomyocytes obtained through this method should be thoroughly discussed.

2. Can you provide a detailed explanation of how you obtained a quality-controlled stem cell bank? What are your standards, and are there corresponding references? Meanwhile, are the cardiomyocytes you obtained also quality-controlled? Have they been identified, and what are the criteria for identification?

3. For the frozen storage experiments, have the authors investigated the impact of freezing time on cardiomyocyte vitality and metabolism, as only one time point was mentioned?

4. The result that calcium transient amplitude and upstroke velocity are highest in frozen bCMs is very interesting. Can you attempt to discuss the reasons for this phenomenon?

Others:

1. Please correct the text overlap in Figure 3k.

2. What is the difference in metabolic capacity of bCMs between fresh and post-thaw recovery in Figure 3r?

3. The bioreactor method and spinner flask method are similar to previously published

4. The yield efficiency is similar to previously published

5. Bunches of qRT-PCR results, authors shall show protein expression levels to convince they are more mature in 3D method than monolayer differentiated CMs.

6. Issues: Fig 3a, f and j shall show a group of cells for each time-point instead of one, also shall show the percentage of cells showed rod or round morphology.

7. Since authors already did EP study, what's the percentage of cells showed ventricular-like, atrial-like, and nodal cell action potentials. Author these data in Figure 6 is interesting, how reproducibly can obtain this kind morphology structure of organoids? Authors shall show more organoids and sectioned staining to convince this is robustly reproduced using their method.

8. Authors shall measure single CM contractile force to compare the 3D and monolayer differentiated CM contractility.

Reviewer #2 (Remarks to the Author):

Most of the concerns have been addressed in the revision. It remains slightly unfortunate that the functional analysis was essentially done in one hiPSC line, albeit different subclones, as we know that functional properties do vary eg the earliest hiPSC-cardiomyocytes marketed by CDI had unusually long action potentials. However, the authors have included the bioreactor runs with more lines also demonstrated their point and also used stirred bioreactors, which are more widely available at lower cost. The statistical analysis has also been improved by more replicates largely where necessary Overall, likely a useful method for others to use. One minor point that still has not been properly addressed and I should have been more specific:

2. Please be consistent in use of hiPSC-CM versus iPSC-CM. We adopted iPSC-CM throughout. They should adopt hiPSC since iPSC conventionally refers to mouse cells.

Reviewer #3 (Remarks to the Author):

The revised manuscript by Prondzynski et al is improved though I still have reservations regarding some of the conclusions made.

Some details related to the criteria for the use of the hPSCs from their MCBs is still lacking. For example was the number of passages following thawing to perform differentiations restricted? how regularly was the karyotype assessed? To that end, it is unclear why the example karyotype shown in Sup Fig 1 is “normalised” to a control, and not an absolute count shown. The authors indicate that use of MCBs contributed to the reproducibility of the bioreactor runs (lines 110, 118-119) but the requirements are not clearly defined.

It is clearer now the correlation of SSEA4% and EB size to bioreactor success from the figure included in the response. From this figure it is evident that these requirements are not mutually exclusive and would encourage the authors to include this figure in the manuscript.

The rationale and approach for how mCM samples were selected for qPCR analysis in Fig 1 is still unclear. They stated that sampling was “randomised” for each monolayer batch, but how this was performed is not described. If it was user-decided there is an inherent bias as the morphology of the cells within the well already gives a clear indication of whether a well has efficiently differentiated before sampling. But the authors also state in the manuscript that the bCMs and mCMs were differentiated in parallel for comparison (line 377), in which case selection of batches is not randomised. However also in figure 1, four bCM batches are shown but only 3 mCM batches.

The new data regarding spinner flask CMs is a nice addition to the paper in particular due to the lower costs involved for a laboratory to set up this technology. Indeed the authors comment in their rebuttal one of the main reasons they are unable to evaluate other bioreactor systems is due to the initial cost for establishing the setup (>\$150K). This should also be acknowledged in the manuscript when comparing the costs of generating CMs by bioreactor vs monolayer as this is a significant investment and cannot be ignored when comparing production costs.

The data and evidence the authors provide regarding the bulk production of cardiac organoids is weak though. It appears self-organisation is based on the presence of cyst/cavities within the bCOs. There is a lack of data demonstrating self organisation within these structures, both during the differentiation and within the final structures. For example, both EC and fibroblast populations are identified by scRNAseq, but the localisation of these cells within the bCOs is not shown.

Reviewer #4 (Remarks to the Author):

Efficient cardiomyocyte differentiation from human PSCs is well established. Variable outcome results typically from poorly controlled or low-quality starting materials and lack of experience. As to the cell material, it is or should be a standard in the field to work with well characterized cell banks. Experienced labs have used numerous iPSC lines with their individually preferred protocols with robust success rates and similar outcome as reported by Prondzynski and colleagues. A preference for one or the other is primarily based on experience and the question to be addressed.

The study by Prondzynski and colleagues is without doubt carefully done and contains plenty of interesting data derived from their preferred differentiation protocol and secondary assays. I doubt that the findings can be generalized or will convince experts to adapt their protocols. Whether the outlined protocol will inform or enable inexperienced users is difficult to predict.

A major concern is that novel technological advances and mechanistic insight are lacking. GiWi differentiation of one kind or the other is used by many labs. Spinner and bioreactor cultures are in broad use for many years. The claimed advantage in cryopreserved spinner cultures is, if at all present, incremental. 94% survival after thawing would be excellent if confirmed by non-trypan blue assays. Trypan blue stains are notoriously imprecise and most importantly will not predict cardiomyocyte plating efficiency, which is a more relevant endpoint. The organoid model is an adaption of lineage restricted embryoid body culture models and it remains unclear, whether or not such models will be of any specific use.

Taken together, this is a protocol paper of high quality. I doubt that the reported data will convince experts to change their protocols. I am also not convinced that inexperienced users should be left with the impression that the reported protocol is superior to others.

RESPONSES TO REVIEWERS

We were pleased to see in the decision letter that the Editors were satisfied with the level of novelty of our revised manuscript, but requested additional revision based on reviewer feedback. We thank the reviewers for their careful review and constructive comments. In this second revision we have addressed the reviewer comment on the first revision with new data, analyses, and changes to the manuscript.

REVIEWER COMMENTS

Reviewer #1 (Remarks to the Author):

The paper introduces an optimized method for producing a large quantity of high-purity hiPSC-induced cardiomyocytes. While various methods have been previously discussed, this approach seems reliable and applicable across multiple hiPSC lines, yielding at least equivalent or superior functional characteristics in derived hiPSC-CMs compared to standard monolayer differentiation. This is encouraging, but there are still some issues to address.

Main concerns:

1. I remain confused about the author's response to the issue of low innovativeness in the rebuttal letter. I am not particularly clear on whether the author intends to convey the enhancement of cardiomyocyte differentiation efficiency and quality by the bioreactor or the subsequent rotating flask suspension system. The author appears to emphasize both products, each with its advantages and disadvantages.

Both suspension culture systems (bioreactor and spinner flask) efficiently generate large numbers of iPSC-CMs with excellent functional characteristics that are superior to standard monolayer-differentiated iPSC-CMs. We found that the bioreactor cells had measurably more mature properties than spinner flask cells; on the other hand, the spinner flask system is less cumbersome and the hardware is far less expensive.

The author acknowledges that the promotion of cardiomyocyte differentiation by the bioreactor has been reported before. Therefore, the novelty of this study, as suggested by the author, lies in the detailed functional identification of the generated cardiomyocytes, which, according to the author, was not meticulous in previous research. However, I believe this is not the innovation point. The highlight of this paper lies in achieving similar effects as adding human dermal fibroblasts with the addition of very few growth factors. The author could emphasize this point in the manuscript and clarify the improvements made with the bioreactor, which are not mentioned in the text.

The bioreactor cells had excellent contractile performance, equivalent to adding dermal fibroblasts to EHTs, as noted by the reviewer. Because functional properties of other bioreactor-derived cardiomyocytes were not previously measured, it is difficult to say if our bioreactor protocol made improvements that resulted in a gain in contractile performance, or if this property had been missed in other bioreactor-derived cells. The striking effect that differentiation protocols have on cell function may not be surprising, yet at the same time efforts to refine differentiation protocols did not perform in-depth

functional characterization of differentiation products and instead focused on cardiomyocyte yield.

We have revised the discussion of the manuscript to (1) include the gain in contractile performance that was comparable to adding human fibroblasts; and (2) to emphasize the link between differentiation protocol and functional characterization, and reiterate the need to functionally characterize products of revised differentiation protocols.

In the subsequent discussion of the rotating flask suspension system, the paper mentions that although it is not as perfect as the bioreactor, it can achieve higher throughput of cardiomyocytes. In fact, the rotating flask suspension system has long been proposed (such as PMID: 26318718). Therefore, I hope the author can reorganize the writing to clearly state which system they believe is better for obtaining highly efficient and homogeneous cardiomyocytes and which is more suitable for widespread application. Once you have determined which one, the functional identification of the cardiomyocytes obtained through this method should be thoroughly discussed.

There are many factors to consider so it is not possible to state that one solution is better than the other. If only considering functional characteristics of cells, then the bioreactor is the preferable system. However, the functional gain is relatively small and the price is high, too high for many labs. In this case, the spinner flasks are an excellent compromise. Furthermore, the spinner flask is more amenable to scale-up than the bioreactor.

2. Can you provide a detailed explanation of how you obtained a quality-controlled stem cell bank? What are your standards, and are there corresponding references? Meanwhile, are the cardiomyocytes you obtained also quality-controlled? Have they been identified, and what are the criteria for identification?

The stem cell bank was described in detail previously (PMID: 32956561). We did not further innovate on the stem cell bank, although it is important for reproducible results. We refer readers to the prior publications about stem cell banks. In the revised manuscript, we added information about our implementation of the master cell bank to the methods.

Regarding the iPSC-CMs, the primary quality control is the yield and the percentage of TNNT2+ cells. Additionally, we evaluated viability and plating efficiency of cryopreserved bCMs and mCMs showing comparable results to commercially available iPSC-CMs. The cells generally had reproducible functional properties even though we did not quality control cells using functional parameters.

3. For the frozen storage experiments, have the authors investigated the impact of freezing time on cardiomyocyte vitality and metabolism, as only one time point was mentioned?

We did not systematically investigate the effect of time in cryostorage on cardiomyocyte vitality and metabolism. However, we have not observed differences in cryorecovery between cells frozen for days to over 3 years.

For the revision, we analyzed the effect of freezing on mitochondrial function using the Seahorses mitochondrial function test. We found no statistical differences between

cryopreserved or fresh bCMs. Fresh mCMs showed significantly higher basal and maximum respiration compared to cryopreserved mCMs. However, basal and maximum respiration as well as ATP production remained statistically lower in fresh mCMs when compared to cryopreserved or fresh bCMs.

4. The result that calcium transient amplitude and upstroke velocity are highest in frozen bCMs is very interesting. Can you attempt to discuss the reasons for this phenomenon?

A similar question was raised in the first revision of this manuscript. In our discussion we mention a previous publication by Van den Brink et al. (PMID: 31945612) showing more ventricular subtypes in cryopreserved vs. freshly plated hiPSC-CMs and proposing an enrichment of ventricular hiPSC-CMs due to cryopreservation. This is in line with our findings as cryo-recovered bCMs displayed features of more mature Ca²⁺ transients and action potentials than fresh bCMs.

Others:

1. Please correct the text overlap in Figure 3k.

Corrected

2. What is the difference in metabolic capacity of bCMs between fresh and post-thaw recovery in Figure 3r?

For the revision, we analyzed the effect of freezing on mitochondrial function using the Seahorses mitochondrial function test. We found no statistical differences between cryopreserved or fresh bCMs. Fresh mCMs showed significantly higher basal and maximum respiration compared to cryopreserved mCMs. However, basal and maximum respiration as well as ATP production remained statistically lower in fresh mCMs when compared to cryopreserved or fresh bCMs.

3. The bioreactor method and spinner flask method are similar to previously published

The bioreactor and spinner flask protocols drew on prior publications. However, prior publications did not functionally characterize the differentiation products. Therefore researchers were unable to decide whether or not to adopt these protocols because the functional properties of the resulting cells were not known. Here we show that differentiation method strongly affects functional properties, and that the suspension culture approaches yield cells with improved functional properties, in addition to greater number and reproducibility. Additionally, we summarize and compare our differentiation protocol to previously published protocols in Table S1.

4. The yield efficiency is similar to previously published.

We listed the yield of prior protocols compared to this publication in Table S1. This analysis shows that the yields reported in this manuscript exceed prior publications.

5. Bunches of qRT-PCR results, authors shall show protein expression levels to convince they are more mature in 3D method than monolayer differentiated CMs.

We analyzed protein levels by western blotting (Fig. S1g). We found statistically higher TNNT2 protein levels in bCMs when compared to mCMs.

6. Issues: Fig 3a, f and j shall show a group of cells for each time-point instead of one, also shall show the percentage of cells showed rod or round morphology.

In Fig 3a, we provided quantitative data of many cells to give an overview of the shape of the hiPSC-CMs, with the image as a representative example. The circularity index as a quantitative measure of roundness. In the revised manuscript, we added thresholded circularity index to show the percentage of cells with rod or round morphology (Fig S3a).

For the patterned hiPSC-CMs, low power images are shown in Fig. S3d. We provide unbiased quantitative analysis of the images to objectively demonstrate the features illustrated in the images (Fig. 3h, k and Fig. S3e-f).

7. Since authors already did EP study, what's the percentage of cells showed ventricular-like, atrial-like, and nodal cell action potentials. Author these data in Figure 6 is interesting, how reproducibly can obtain this kind morphology structure of organoids? Authors shall show more organoids and sectioned staining to convince this is robustly reproduced using their method.

We performed voltage measurements using a high-content imaging device by staining the cells with FluoVolt. Du et al. (PMID 25564842) previously showed that optically recorded APs do not reliably predict cardiac chamber specificity.

We revised Fig. 6 and Fig. S10 by showing more sectioned and stained organoids to show that the morphology can be reproducibly generated in suspension-cultured organoids..

8. Authors shall measure single CM contractile force to compare the 3D and monolayer differentiated CM contractility.

EHTs are a well-established method to measure contractile force of iPSC-CMs (PMID: 28492526). We did not further measure single CM contractile force as the methods to make these measurements are not currently established in our lab and the measurement of single CM contractile force is peripheral to this manuscript.

Reviewer #2 (Remarks to the Author):

Most of the concerns have been addressed in the revision. It remains slightly unfortunate that the functional analysis was essentially done in one hiPSC line, albeit different subclones, as we know that functional properties do vary eg the earliest hiPSC-cardiomyocytes marketed by CDI had unusually long action potentials. However, the authors have included the bioreactor runs with more lines also demonstrated their point and also used stirred bioreactors, which are more widely available at lower cost. The statistical analysis has also been improved by more replicates largely where necessary Overall, likely a useful method for others to use. One minor point that still has not been properly addressed and I should have been more specific:

2. Please be consistent in use of hiPSC-CM versus iPSC-CM. We adopted iPSC-CM throughout.

They should adopt hiPSC since iPSC conventionally refers to mouse cells.

We thank the reviewer for the supportive comments. We revised the manuscript to use hiPSC-CM as recommended by the reviewer, although we note that many investigators use iPSC-CM for human cells.

Reviewer #3 (Remarks to the Author):

The revised manuscript by Prondzynski et al is improved though I still have reservations regarding some of the conclusions made.

Some details related to the criteria for the use of the hPSCs from their MCBs is still lacking. For example was the number of passages following thawing to perform differentiations restricted? how regularly was the karyotype assessed? To that end, it is unclear why the example karyotype shown in Sup Fig 1 is “normalised” to a control, and not an absolute count shown. The authors indicate that use of MCBs contributed to the reproducibility of the bioreactor runs (lines 110, 118-119) but the requirements are not clearly defined.

We adopted MCB procedures that were described previously. Since we did not innovate on MCBs, we did not include much text about them. In light of comments from Reviewers 1 and 3, we added a paragraph on MCBs to the revised Methods, including a description of karyotyping:

Master cell banks (MCBs) were established as described previously²⁵. In short, hiPSCs were grown and massively expanded in T80 cell culture flasks (Life Technologies, #178905; see details below). Expanded hiPSCs were dissociated and all resulting cells were frozen in mFreSR™ (STEMCELL Technologies, #05854) at 3 million hiPSC per cryovial, whereby 4x T80 flasks yield ~ 60 million hiPSCs. These cells were tested for karyotype abnormalities, genotyped and screened for mycoplasma contamination. Massive expansion of hiPSCs derived from the MCB was repeated to generate a working cell bank. For subsequent cardiac differentiation and functional analysis, solely cells derived from MCBs and working cell banks were used to increase reproducibility between independent differentiation batches.

The example karyotype used a digital karyotype method (Nanostring) in which the signal of each probe is normalized to the overall genome average of a sample with a known karyotype. Detailed data processing for this karyotyping method are described here: https://nanostring.com/wp-content/uploads/MAN-C0014-02_nCounter_CNV_Data_Analysis_Guidelines.pdf

It is clearer now the correlation of SSEA4% and EB size to bioreactor success from the figure included in the response. From this figure it is evident that these requirements are not mutually exclusive and would encourage the authors to include this figure in the manuscript.

We added the figure to the supplementary figure.

The rationale and approach for how mCM samples were selected for qPCR analysis in Fig 1 is still unclear. They stated that sampling was “randomised” for each monolayer batch, but how this was performed is not described. If it was user-decided there is an inherent bias as the morphology of the cells within the well already gives a clear indication of whether a well has efficiently differentiated before sampling. But the authors also state in the manuscript that the bCMs and mCMs were differentiated in parallel for comparison (line 377), in which case selection of batches is not randomised. However also in figure 1, four bCM batches are shown but only 3 mCM batches.

Samples for molecular analysis were picked in a random fashion without any visual confirmation of beating, therefore without user bias. This also explains why some samples show very little cardiomyocyte-specific gene or protein expression, since all 12 wells in a multidish well plate show high variability in hiPSC-CM content (Movie S4). However, before mCMs dissociation and freezing, wells were visually evaluated for spontaneous contractions and only wells containing >60% hiPSC-CMs were subsequently dissociated and further processed for functional experiments. This step was introduced, since unbiased selection at this point would have significantly reduced the CM content in mCMs batches making functional comparison to bCMs more difficult.

The new data regarding spinner flask CMs is a nice addition to the paper in particular due to the lower costs involved for a laboratory to set up this technology. Indeed the authors comment in their rebuttal one of the main reasons they are unable to evaluate other bioreactor systems is due to the initial cost for establishing the setup (>\$150K). This should also be acknowledged in the manuscript when comparing the costs of generating CMs by bioreactor vs monolayer as this is a significant investment and cannot be ignored when comparing production costs.

We added hardware costs to Supplementary Table 3.

The data and evidence the authors provide regarding the bulk production of cardiac organoids is weak though. It appears self-organisation is based on the presence of cyst/cavities within the bCOs. There is a lack of data demonstrating self organisation within these structures, both during the differentiation and within the final structures. For example, both EC and fibroblast populations are identified by scRNAseq, but the localisation of these cells within the bCOs is not shown.

We provide additional data on staining the bCOs with markers for ECs and fibroblasts in Fig. S10. The structures are self-organizing in that they spontaneously form spheres in which myocardial walls bound a central cavity, which has one or more septae.

Reviewer #4 (Remarks to the Author):

Efficient cardiomyocyte differentiation from human PSCs is well established. Variable outcome results typically from poorly controlled or low-quality starting materials and lack of experience. As to the cell material, it is or should be a standard in the field to work with well characterized cell banks. Experienced labs have used numerous iPSC lines with their individually preferred protocols with robust success rates and similar

outcome as reported by Prondzynski and colleagues. A preference for one or the other is primarily based on experience and the question to be addressed.

The study by Prondzynski and colleagues is without doubt carefully done and contains plenty of interesting data derived from their preferred differentiation protocol and secondary assays. I doubt that the findings can be generalized or will convince experts to adapt their protocols. Whether the outlined protocol will inform or enable inexperienced users is difficult to predict.

One reason that protocol selection has been driven by individual experience and preference is that the functional data has not been available to make more scientifically driven choices. In our opinion, providing the functional data is critical to allow more rational comparison and selection between protocols. We have already shared the protocol with several labs in the US, Canada, and Europe, and have hosted investigators from several labs at our lab to learn the protocol. This suggests to us that there is tremendous interest.

A major concern is that novel technological advances and mechanistic insight are lacking. GiWi differentiation of one kind or the other is used by many labs. Spinner and bioreactor cultures are in broad use for many years. The claimed advantage in cryopreserved spinner cultures is, if at all present, incremental. 94% survival after thawing would be excellent if confirmed by non-trypan blue assays. Trypan blue stains are notoriously imprecise and most importantly will not predict cardiomyocyte plating efficiency, which is a more relevant endpoint. The organoid model is an adaption of lineage restricted embryoid body culture models and it remains unclear, whether or not such models will be of any specific use.

We believe that the protocol coupled with deep functional characterization of the produced cells is an important advance. Comparison to other iPSC-CM differentiation protocols is provided in Table S1 and shows the improvement in yield. In most cases, comparison of cell functional qualities was not possible due to lack of measurement of these parameters in prior studies.

Regarding survival assays, the trypan blue assay is widely used in academia and industry and therefore valuable as a means to compare between studies. In the revised manuscript, we provide data on plating efficiency, which is comparable to the plating efficiency of commercially available iPSC-CMs.

Taken together, this is a protocol paper of high quality. I doubt that the reported data will convince experts to change their protocols. I am also not convinced that inexperienced users should be left with the impression that the reported protocol is superior to others.

We provide a systematic comparison to other iPSC-CM protocols in Table S1. The yield is higher than prior studies. We provide data on key functional parameters and show that these change depending on the differentiation protocol. Unfortunately other protocols focused on yield and did not measure functional properties, so it is not possible to objectively compare the cells' functional properties.

REVIEWERS' COMMENTS

Reviewer #1 (Remarks to the Author):

The manuscript is significantly improved. I have following suggestions to further improve the manuscript:

- Fig. 1g: Authors showed overall image of bio-reactor differentiated hiPSC-CMs. What about apoptosis of CM in bCMs at day 30 or even longer time culture.
- Fig 4i: did authors use the same hiPSC line for generating bCM and mCM and used them for manufacturing EHT.
- Fig. 4J: The data of sarcomere length ranged from 0.5 -2.5 μm . Electron microscopy could be used to quantify sarcomere length.
- Authors indicated that Bioreactor-differentiated CMs showed predominantly ventricular identity. The results are mainly based on RT-PCR, or RNA seq results. The demonstration requires the staining, flow results, and EP studies .

Reviewer #3 (Remarks to the Author):

My comments have been addressed.

REVIEWERS' COMMENTS

Reviewer #1 (Remarks to the Author):

The manuscript is significantly improved. I have following suggestions to further improve the manuscript:

- Fig. 1g: Authors showed overall image of bio-reactor differentiated hiPSC-CMs. What about apoptosis of CM in bCMs at day 30 or even longer time culture. Do you have an apoptosis measurement

We analyzed nuclear morphology in micropatterned frozen bCMs and mCMs over the duration of 7 days (Fig. S4) after differentiation (Day 22). We observed a significantly higher fraction of “small round” nuclei in mCMs (6.6-13.8%) when compared to bCMs (0-2.6%). These results are derived using previously published unbiased image analysis

software (PMID: 22905142) in which “small round” nuclei are classified as apoptotic nuclei. Additionally, we validated cell coverage of micropatterned frozen bCMs and mCMs over the duration of 7 days and bCMs better survived plating on the micropatterned substrates, as demonstrated by their markedly higher coverage at all time points compared to mCMs (Fig3g; Fig. S3d).

However, unpatterned iPSC-CMs have very low apoptosis rates. This is supported by staining for apoptotic markers annexin V and caspase 3 in bCMs, mCMs, and sCMs, frozen at day 15 after then cryo-recovered for 7 days (i.e., about 22 days after differentiation; Rebuttal Fig. 1). However, we do not have a measurement of apoptosis at day 30 or longer.

Fig. 1: Representative images of cryopreserved hiPSC-CMs 7 days after plating in 96-well plates stained for actinin 2 (ACTN2), annexin V, caspase 3 (CASP3) and Hoechst 33342 for nuclei. Bar, 500 µm.

• Fig 4i: did authors use the same hiPSC line for generating bCM and mCM and used them for manufacturing EHT.

For all functional comparisons of bCMs, sCMs and mCMs we used our healthy control line WTCCas9, which is derived from the commercially available WTC-11 line. This is now stated in all paragraphs focused on functional comparisons in the revised manuscript.

- Fig. 4J: The data of sarcomere length ranged from 0.5 -2.5 μm . Electron microscopy could be used to quantify sarcomere length.

Although EM can be used to measure sarcomere length, it can also be reliably measured using light microscopy. Indeed light microscopy can sample more sarcomeres and may represent the overall sarcomere length and distribution better than EM.

- Authors indicated that Bioreactor-differentiated CMs showed predominantly ventricular identity. The results are mainly based on RT-PCR, or RNA seq results. The demonstration requires the staining, flow results, and EP studies .

We performed flow cytometry analysis of bCMs 15 days after differentiation and observed that 83.4% hiPSC-CMs stained for ventricular marker myosin light chain 2v (MLC2v; new Fig. S1g). This finding is in line with scRNAseq results. We integrated these data in the revised MS. We additionally performed stainings for MLC2v in bCMs, mCMs and sCMs 7 days after cryo-recovery (Day 22) and observed >70% MLC2v-positive hiPSC-CMs in bCMs and sCMs (Rebuttal Fig. 2).

Fig. 2: Representative images of cryopreserved hiPSC-CMs 7 days after plating in 96-well plates stained for actinin 2 (ACTN2), myosin light chain 2v (MLC2v), myosin light chain 2a (MLC2a) and Hoechst 33342 for nuclei. Bar, 500 μm .

Reviewer #3 (Remarks to the Author):

My comments have been addressed.